# Characterization of the Key Aroma Volatile Compounds in Nine Different Grape Varieties Wine by Headspace Gas Chromatography–Ion Mobility Spectrometry (HS-GC-IMS), Odor Activity Values (OAV) and Sensory Analysis

**DOI:** 10.3390/foods11182767

**Published:** 2022-09-08

**Authors:** Weiyu Cao, Nan Shu, Jinli Wen, Yiming Yang, Yuning Jin, Wenpeng Lu

**Affiliations:** Institute of Special Animal and Plant Sciences of Chinese Academy of Agricultural Sciences, Changchun 130112, China

**Keywords:** wine, HS-GC-IMS, volatile components, aroma-presenting substances, odor activity value, variable importance in projection value

## Abstract

During this study, the physicochemical properties, color, and volatile aroma compounds of the original wines produced from the grape varieties ‘Hassan’, ‘Zuoshaner’, ‘Beibinghong’, ‘Zuoyouhong’, ‘Beta’, ‘Shuanghong’, ‘Zijingganlu’, ‘Cabernet Sauvignon’, and ‘Syrah’ were determined and sensory evaluation was performed. Results indicated that ‘Hassan’ contained the most solids, ‘Zuoshaner’ produced the most total acid, residual sugar, total anthocyanin, and total phenol, and ‘Shuanghong’ produced the most tannin. Calculation of the chroma and hue of the wines according to the CIEL*a*b* parameters revealed that the ‘Cabernet Sauvignon’ wines were the brightest of the nine varieties and that the ‘Zuoshaner’ wines had the greatest red hue and yellow hue and the greatest saturation’. A total of 52 volatile compounds were identified and quantified in nine wine samples by HS-GC-IMS analysis, with the most significant number of species detected being 20 esters, followed by 16 alcohols, 8 aldehydes, four ketones, one terpene, and one furan, with the highest total volatile compound content being ‘Beta’. A total of 14 volatile components with OAV (odor activity value) >1 were calculated using the odor activity value (OAV) of the threshold of the aromatic compound, and the OPLS-DA analysis was performed by orthogonal partial least squares discriminant analysis (OPLS-DA) using the OAV values of the compounds with OAV values >1 as the Y variable. The VIP (Variable Importance in Projection) values of six compounds, ethyl isobutyrate, ethyl hexanoate-D, 2-methylpropanal, ethyl octanoate, ethyl butanoate-D, and Isoamyl acetate-D, were calculated to be higher than one between groups, indicating that these six compounds may influence aroma differences. It is essential to recognize that the results of this study have implications for understanding the quality differences between different varieties of wines and for developing wines that have the characteristics of those varieties.

## 1. Introduction

As the global wine market continues to grow, the question of how to enhance the flavor of the wine is increasingly becoming a hot research topic. Regular consumption of red wine in moderation has been shown to positively affect health, with wine containing phenolic compounds with antioxidant properties [1]. Wine is an alcoholic beverage product obtained by fermentation of fresh grapes or grape juice, with components derived from grape-kernels and the fermentation and aging process [2]. The organoleptic properties of wine are conferred by organoleptically active compounds, mainly polyphenols (coloring), sugars, acids, tannins (taste), and volatile flavor compounds (aroma) [3]. The blend and balance of aromas in a wine determine the quality of the wine and how well the consumer likes it, expresses the style of the wine, and is a significant indicator of the quality of the wine [4]. The composition and content of the substances in different wines vary, and these differences determine the wines’ flavor and quality. The distinctive aromas of wine come from the hundreds of volatile compounds formed in the grapes during the ripening and maturation stages. These volatile compounds include alcohols, esters, acids, aldehydes, ketones, and terpenes [5]. However, not all volatile compounds are responsible for the overall aroma of a bottle of wine. The olfactory impact of these compounds depends on their concentration and the corresponding threshold for identifying the impacted odor [6].

Current research divides wine aromas into three types; The first refers to the aromas in the berries, which are related to the variety of the raw material and the environment in which they are grown and are the most critical factor in determining the type of grape aroma. The second refers to the aromas produced by the berries during fermentation, which has a significant impact on the aroma of the final wine and not only has a direct effect on the aromas but also modifies and provides the basis for varietal and aging aromas. The third component is the aromas produced by the wine during its aging process. Volatile compounds in wine are usually small molecular weight compounds that give wines different odor characteristics, including floral, fruity, woody, herbal, animal, and roasted [7]. Aroma activity values (OAV) and partial least squares discriminant analysis (OPLS-DA) are commonly used to identify characteristic food aroma components [8].

The conventional methods for analyzing volatile substances in food include gas chromatography–mass spectrometry, headspace solid-phase microextraction, and gas chromatography–mass spectrometry [9,10]. An ion mobility spectrometer (IMS) is ultra-sensitive with ultra-high analytical speed. This simple-to-operate instrument can operate at atmospheric pressure and requires no sample preparation steps. IMS is an analytical technique for detecting trace gases and the characterization of chemical ionic substances based on differences in the mobility of gas-phase ions in an electric field [11]. The application of headspace gas chromatography–ion mobility spectrometry (HS-GC-IMS) has been widely reported in recent years for determining volatile components in foodstuffs [12,13,14,15,16]. HS-GC-IMS takes advantage of the separation characteristics of GC and the fast response and high sensitivity of IMS, which can detect a large number of compounds with different chemical groups, including alcohols, aldehydes, aromatics, esters, and ketones, even from the most complex and problematic matrices, such as food and agricultural products [17]. HS-GC-IMS improves the accuracy of qualitative analysis and produces three-dimensional spectra of retention time, drift time, and signal intensity; it has been successfully used to analyze flavor and quality in food products, including the analysis of different species of volatile organic compounds and their metabolites [18,19,20].

Although some research work has been carried out on wine by previous authors, such as studies on the factors affecting wine flavor, from different perspectives such as variety [21,22,23], cultivation techniques [24,25], environment [26,27,28], fermentation process [29], aging process [30,31], and storage methods [32], research on the flavor of wine still needs to be strengthened. Firstly, previous studies have only analyzed the aroma components in wine, failing to identify the key aroma components [33,34,35,36]. Furthermore, there is a lack of sensory evaluation, which is the most subjective and effective way for consumers to judge the quality of the wine. Secondly, many previous studies have reported the application of HS-GC-IMS for detecting volatiles in food and agricultural products [19,37,38,39,40,41]. However, the use of HS-GC-IMS for detecting target compounds in wine samples has been rarely reported, especially the application of HS-GC-IMS for identifying volatile compounds in *Vitis amurensis* wines. The application of HS-GC-IMS to identify volatile compounds in *Vitis amurensis* wines, in particular, has rarely been reported.

In this study, the fundamental physicochemical properties, color, volatile aroma compounds, and the organoleptic properties of the original wine made from nine varieties of grapes harvested in 2021 were determined, and a fingerprint profile of the volatile compounds of the different varieties of wine was established. Moreover, based on the volatile compounds by multivariate statistical analysis quantitative descriptive analysis data, the specific wine aroma characteristics were characterized while combined with principal component analysis, OAV value analysis, and VIP value analysis to screen the key volatile compounds affecting wine aroma and identify the volatile compounds that may affect wine flavor. It provides a theoretical basis for enhancing and improving wine quality, a scientific understanding of the nature of flavor chemistry of aroma characteristics of different wine varieties, and product flavor quality control.

## 2. Materials and Methods

### 2.1. Materials and Reagents

#### 2.1.1. Sample Preparation

The experiment was conducted in September 2021 with nine grape varieties, including ‘Hassan’, ‘Zuoshaner’, ’Beibinghong’, ‘Zuoyouhong’, ‘Beta’, ‘Shuanghong’, ‘Zijingganlu’, ‘Cabernet Sauvignon’, and ‘Syrah’. The sampling site was the National fruit tree germplasm *Vitis amurensis* nursery in Zuojia town, Institute of Special Products, Chinese Academy of Agricultural Sciences, and the sampling period was when the fruit was ripe. The nine varieties were planted from late April to early May at a spacing of 1.0 m × 2.5 m. The entire garden was managed with conventional fertilizer and water. A single-armed hedge frame was used in the resource bed, with a hedge frame spacing of 1.0 m × 2.5 m and a trellis frame spacing of 1.0 m × 3.0 m. A 30 cm in diameter and 30 cm deep planting hole was dug in the center of the planting trench, with the seedlings’ roots naturally and evenly distributed in the hole. We weeded 4–5 times a year and fertilized thrice a year, mainly with organic fertilizer. The varieties were managed uniformly. Sampling was carried out at different locations in the vineyard by selecting fruit ears of each variety that are free from pests and diseases, and free from mold; 10 kg of each variety was taken back to the laboratory on the same day in a thermos for use in the winemaking.

#### 2.1.2. Reagents

Analytical purity: sulfuric acid, sodium chloride, potassium chloride, sodium bicarbonate (Beijing Chemical Factory, Beijing, China); tannic acid (Tianjin Guangfu Fine Chemical Research Institute, Tianjin, China); Folin-Denis reagent (US sigma, St. Louis, MO, USA); anhydrous sodium carbonate (Tianjin Hengxing Chemical Reagent Manufacturing Co., Ltd., Tianjin, China); glacial acetic acid, hydrochloric acid, anhydrous ethanol, sodium hydroxide, phosphoric acid (Beijing Chemical Factory); potassium hydrogen phthalate, anthrone (Sinopharm Chemical Reagent Co., Ltd., Shanghai, China); anhydrous sodium acetate (Shanghai Hutian Chemical Co., Ltd., Shanghai, China); glucose (Guangzhou Jinhuada Chemical Reagent Co., Ltd., Guangzhou, China).

Chromatographic purity: methanol (TEDIA Reagents, Fairfield, OH, USA); succinic acid, fumaric acid, malic acid, citric acid anhydrous (Shanghai Yuanye Biotechnology Co., Ltd., Shanghai, China); 4-methyl-2-pentanol (Shanghai Lianshuo Biotechnology Co., Ltd., Shanghai, China); glacial acetic acid, tartaric acid, lactic acid (Tianjin Fine Chemical Research Institute, Tianjin, China)

Fermentation auxiliaries: CEC01 active dry yeast (Angel Yeast Co., Ltd., Hubei, China); potassium metabisulphite (Yantai Dibs Homebrewer Co., Ltd., Yantai, China).

### 2.2. Instrumentation and Equipment

High-performance liquid chromatograph (Agilent Technologies, Waldbronn, Germany); FlavourSpec^®^ flavor analyzer (G.A.S.); electronic balance-purchased from Sartorius Scientific Instruments (Beijing, China); digital vernier calipers—purchased from Seda Tools Co. (Shanghai, China); Wine Refractometer (ATAGO), CJJ-931 Duplex Magnetic Heating Stirrer (Jiangsu Jintan Jincheng Guosheng Experimental Instrument Factory, Jiangsu, China); HWS-12 Electric Constant Temperature Water Bath, KQ-300E Ultrasonic Cleaner, Snowflake Ice Maker (Beijing Changliu Scientific Instrument Co., Ltd., Beijing, China), FA1004B Electronic Balance (Shanghai Yue Ping Scientific Instrument Co., Ltd., Shanghai, China), DHG- 9240 (Shanghai Yiheng Scientific Instruments Co., Ltd., Shanghai, China), WAX column (RESTEK, Bellefonte, PA, USA).

### 2.3. Methodology

#### 2.3.1. Winemaking

After harvesting, the grapes were destemmed, grape crushed by manual destemming and crushing, and fermented at room temperature (25 °C). The fermenting tank was made from 304 stainless steel thermostatic fermentation vats from Tiburth, with a volume of 12 L. Three sets of replicated winemaking experiments for each variety, and each fermenter was filled with around 10 L of crushed grapes. During fermentation, the fermenting tank was tightly closed, and an exhaust valve was used to ensure that the gas produced during fermentation was discharged smoothly. The first fermentation lasted seven days, and by testing the sugar and alcohol content during the fermentation period, the total sugar content of each variety of wine stopped decreasing at the end of the first fermentation and remained stable, and no bubbles were produced in the fermenter, while the alcohol content reached a certain concentration. The second fermentation for one month was mainly to check the difference in physical and chemical indexes of each variety at one month of aging. Fermentation temperature for the second fermentation was between 18 and 20 °C. During the second fermentation, the indexes of each variety were already stable.

The yeast used for the fermentation was CEC01 active dry wine yeast from Angel’s yeast. The yeast was added at 250 mg/Kg, and the SO2 was added at 60 mg-L-1.

Fermentation flow chart for wine (Figure 1):

#### 2.3.2. Testing the Basic Physical and Chemical Properties of Raw Wine Grapes

The soluble solids must be determined by handheld refractometer, and the titratable acid content of wine was determined by the indicator method according to GB/T 15,038–2006 General Analysis Method of Wine and Fruit Wine. The alcohol content was determined by the alcohol meter method according to GB/T 15,038–2006 General Analysis Method of Wine and Fruit Wine. Anthrone and sulfuric acid colorimetry were used to determine the total sugar content in grapes wine, and the standard curve was prepared with standard glucose solution; the Folin-Denis reagent method was used to determine tannin content in grapes Juice, and the standard curve was created with different tannin concentrations. It was reacted with phosphomolybdic acid in sodium carbonate solution to form the blue compound after being soaked in water at 85 °C for three hours. The absorbance value was measured at 740 nm.; total anthocyanin content in grape juice was determined by the pH difference method by reacting anthocyanins with potassium chloride buffer (0.025 M, pH = 1) and acetic acid buffer (0.4 M, pH = 4.5), then calculating differences at 520 nm and 700 nm. Total phenol content: Folin–Ciocalteu colorimetric method [42]. Dry extraction content: refer to the dry extraction test method of the national standard (GB/T 15,038–2006).

#### 2.3.3. Colorimetry

The colorimetric analysis was based on the CIEL*a*b* colorimetric standard, and the color characteristics of the wine samples were measured spatially using a Lambda 365 UV–Vis spectrophotometer with continuous scanning (400–700 nm) and distilled water as a blank control group. *L**, *a**, *b**, Cab*, hab*, and Δ*Eab** were calculated based on the four absorbance values, L value indicates brightness, *a** = red-green deviation, *b** = blue-yellow deviation, hab* indicates hue angle, Cab* indicates red grape color index, and Δ*Eab** indicates the total color difference
(Δ*E_ab_**)^2^ = (*L** − *L*_0_*)^2^ + (*a** − *a*_0_*)^2^ + (*b** − *b*_0_*)^2^

#### 2.3.4. Determination of Organic Acid Content

The organic acids were detected using high-performance liquid chromatography (HPLC), following the previously published literature as a reference, under the following conditions: aqueous phosphoric acid solution at pH = 2.3, methanol as mobile phase, and the test conditions were: a C18-XT column (4.6 × 250 × 5) at a column temperature of 25 °C and a set flow rate of 0.4 mL/min [43]. The standard curves for the six organic acids tested were as Table 1.

#### 2.3.5. Quantification of Volatile Compounds in Wine by Headspace-Gas Chromatography–Ion Mobility Spectrometry (HS-GC-IMS)

Headspace-gas chromatography–ion mobility spectrometry (HS-GC-IMS) was used for the determination of volatiles in wine. The instrument used in the experiment was a G.A.S. FlavourSpec^®^ flavor analyzer. Briefly, 1 mL of the sample was taken in a 20 mL headspace vial, 10 µL of 20 ppm 4-methyl-2-pentanol was added, incubated at 60 °C for 15 min, and then injected into the sample. Chromatographic conditions: The column was a WAX column (15 m × 0.53 mm, 1 µm), column temperature 60 °C, carrier gas N2, and IMS temperature 45 °C. The conditions for the automatic headspace injection were as follows: injection volume 100 μL, incubation time 10 min, incubation temperature 60 °C, injection needle temperature 65 °C, and incubation speed 500 rpm; The analysis was carried out using 4-methyl-2-pentanol as an internal standard at a concentration of 198 ppb. The volume of the signal peak was 493.34 and the intensity of each peak was approximately 0.401 ppb. And Table 2 shows the gas chromatography conditions. 

Quantitative calculation formula:$$Ci=\frac{Cis*Ai}{Ais}$$

*Ci* is the calculated mass concentration of any component in µg/L, *Cis* is the mass concentration of the internal standard used in µg/L, and *A*i/AIS is the volume ratio of any signal peak to the signal peak of the internal standard. The NIST database and IMS database are built into the software for the qualitative analysis of the substances. 

#### 2.3.6. Odor Activity Value (OAV) Calculation

The contribution of the overall aroma of wine is evaluated using the odor activity value (OAV). The OAV is calculated by dividing the concentration of volatile compounds by the odor threshold (OT). Volatile compounds with an OAV > 1 are considered to be aromatically active and play an essential role in developing the wine’s aromatic profile.

### 2.4. Sensory Evaluation

A quantitative descriptive analysis (QDA) of the wines was carried out by a trained sensory panel of 13 sommeliers (8 women and 5 men, aged from 22 to 52 years, average 34 years). The experts were recruited based on their motivation and availability and had been trained according to national standards ISO 6658 and ISO 8586 prior to the sensory evaluation. The experts discussed the aromatic composition of the wines in depth over three preliminary meetings (2 h each) until they all agreed on the degree of aromaticity. Wine odors were quantified using six sensory descriptors (floral, fruity, botanical and herbal, fermented, tarry, and confectionery) for sensory characteristics descriptors, based on definitions in the published literature, according to the national standard GB 15,038–2006 and in conjunction with references [44,45], taking into account the results of the discussions. Samples were labeled with three numbers and presented to the tasters in random order. Panel members were asked to rate the intensity of each attribute on a 10-point scale, where a score of 10 indicated the highest intensity and a score of 0 indicated absence. Each sample was assessed in triplicate, and the mean of each sample was expressed by the average of the three scores based on a ten-point scale.

### 2.5. Data Processing

The trial data were statistically collated using Excel 2010, and an analysis of variance (ANOVA) was performed by SPSS (version 22.0, IBM, Armonk, NY, USA). Statistical analysis was performed to check for significant differences in individual results for the trial data, and all data were expressed as mean ± standard deviation. Differences between the two groups were considered significant at *p* < 0.05. Simca software was used for OPLS-DA and VIP value analysis; the GC-IMS assay was smoothed and denoised with Savitzky Golay. The migration time was normalized by setting the RIP position to 1, i.e., dividing the actual migration time by the RIP peak exit time to obtain the approximate migration time. Direct comparison of spectral differences between samples was performed using the Reporter plug-in and comparison of fingerprint profiles using the Gallery Plot plug-in for visual and quantitative comparison of volatile organic compound differences between different samples. Heat map analysis and correlation analysis were performed using the OmicShare tools, a free online platform for data analysis (https://www.omicshare.com/tools (accessed on 27 August 2022)).

## 3. Results and Analysis

### 3.1. Basic Physico-Chemical Indicators of Different Grape Varieties of Raw Wine

As a whole, there were significant differences between the fundamental physicochemical indicators of the wine samples of each variety (Table 3). The residual sugar and solids content of the wines determine the type of wine, which can be classified as dry, semi-dry, semi-sweet, or sweet according to the sugar content [46]. The phenolics are factors that produce bitterness and astringency [47]. The main coloring compound in red wines is anthocyanin [48,49], whose composition and content influence the color characteristics of the wine. As can be seen in the table above, ‘Hassan’ has the highest solid content at 9.4 g/L, which differs significantly (*p* < 0.05) from the other eight varieties of wine samples; ‘Zuoshaner’ has the highest total acid, residual sugar, total anthocyanin, and total phenol content at 16.25 g/L, 5.7 g/L, 1477.85 mg/L, and 3.56 g/L, respectively, which differs significantly (*p* < 0.05) from the other varieties. Tannin is the source of bitterness and astringency in fruit wines. It is also an essential component of the backbone of fruit wines and has a very positive effect on color stabilization, prevention of oxidation, and removal of off-flavors [50]. ‘Shuanghong’ had the highest tannin content of 3.64 g/L, which was significantly different from the other varieties (*p* < 0.05), while the lowest tannin content was 1.3 g/L in ‘Syrah’. The results of the data show that the total anthocyanins, total phenols, and tannins of ‘Hassan’, ‘Zuoshaner’, ‘Beibinghong’, ‘Zuoyouhong’, ‘Shuanghong’, and ‘Zijngganlu’ are significantly higher than those of the American variety of grape ‘Beta’ and the Eurasian varieties of ‘Cabernet Sauvignon’ and ‘Syrah’. This is also in line with the results of previous studies [51,52]. Because of their small size and thick skin, *Vitis amurensis* is very rich in anthocyanins, tannins, and total phenols and produce wines with the right taste. It is also because of the richness of the anthocyanin content of the mountain grape variety that the original wine is darker than the Eurasian and American varieties. The dry extract content is an essential indicator of wine quality, mainly determined by the variety and the age of the wine [53], and the analysis of the dry leachate content in wine can tell whether the wine is adulterated with water, alcohol, etc. According to China’s national standards, the dry leachate of red wine should not be less than 18.0 g/L, the dry extract content of the nine tested varieties met the winemaking standards; the alcohol content of the nine raw grape wine samples ranged from 11° to 13°.

### 3.2. Comparison of the Chromaticity of Different Varieties of Raw Wine Grapes

#### 3.2.1. Variation in Absorbance in the Visible Band of the Samples

The absorbance of the nine wine samples increased continuously in the 400–520 nm band (Figure 2), reaching a maximum of around 520 nm, and decreased significantly in the 520–700 nm band, gradually converging to zero as the wavelength approached 700 nm.

#### 3.2.2. Sample CIELab Parameters

The chromaticity of wine is an essential criterion for evaluating the quality of its appearance, and the degree of oxidation and quality of a bottle of wine can be judged by its chromaticity and hue. New red wines have a purplish-red hue due to anthocyanins. As they mature, the blue-violet hue disappears as the anthocyanins combine with other substances. The wine gradually gains a yellowish hue due to polymeric tannins, with the hue gradually changing from initially purplish-red to tile-red or brick-red. The color tone of the wine is also a critical factor in the marketing of wine [54]. As can be seen from Table 4, the L* values of all nine varieties of raw grape wine samples were high, ranging from 41.65 to 84.65, indicating that all nine varieties of wine have a good luster. The largest L* was ‘Cabernet Sauvignon’ at 84.65, with the brightest color of the wine, followed by ‘Syrah’ and ‘Beta’, and the smallest L* was ‘Zuoshaner’ at 41.65, with the darkest color.

Chroma a* values indicate the red hue of the wines, with the largest a* value being 167.48 for ‘Zuoshaner’, followed by ‘Shuanghong’ and ‘Zuoyouhong’, and the smallest being 17.5 for ‘Cabernet Sauvignon’; chroma b* values indicate the yellow hue of the wines, with nine samples ranging from 1.58 to 21.76, the largest being ‘Zuoshaner’ and the smallest being ‘Syrah’; saturation cab* is a combination of a* and b*, indicating the color of the wines The greatest saturation was for ‘Zuoshaner’, followed by ‘Zouyouhong’ and ‘Shuanghong’, the smallest was for ‘Cabernet Sauvignon’ at 17.93. The saturation and chroma values for the nine varieties were close to each other, which is characteristic of young wines; the hue angle of the nine samples ranged from 4.02 to 23.52, all close to zero, i.e., all close to the purple-red hue. The largest is ‘Zijingganlu’, which is less red than the other eight varieties, and the smallest is ‘Syrah’, which is the closest to purplish red.

The nine wine samples were analyzed for the color difference using Cabernet Sauvignon, which had the highest L* value, as the base value. They were calculated according to the range of color difference units (National Bureau of Standards Unit (NBS)) given by the CIE1976Lab color space system to describe the degree of color difference between the wine samples [55]. As seen from the values in the table, there is a significant color difference between Cabernet Sauvignon and the other eight varieties, with substantial differences.

### 3.3. Comparison of Organic Acids in Wine Samples of Different Varieties

The type of organic acid affects the acidity and, therefore, the taste of the wine. The organic acid content varies from variety to variety (Table 5). Organic acids are an essential part of the structure of the wine, with tartaric acid, the characteristic organic acid of wine with a sour and astringent taste [56], being the most abundant of the six organic acids. ‘Zuoshaner’ had the highest tartaric acid content at 7.99 g/L, significantly higher than the other eight varieties, followed by ‘Zuoyouhong’, with the lowest tartaric acid content being ‘Cabernet Sauvignon’ at 3.16 g/L. Malic acid is present in high levels in grapes and at the end of alcoholic fermentation. It gradually decreases after MLF, a crucial secondary fermentation in most wine production, usually carried out by lactic acid bacteria after the completion of alcoholic fermentation, converting sharp L-malic acid into soft L-lactic acid, which improves microbial stability through residual lactic acid bacteria nutrients and can promote sensory regulation of wine in the secondary metabolism of lactic acid bacteria [57]. The highest malic acid content was in ‘Beta’ at 9.51 g/L, followed by ‘Zuoshaner’ and ‘Hassan’, while the lowest malic acid content was in ‘Syrah’ at 2.03 g/L. The highest levels of lactic acid were found in ‘Hassan’ and ‘Zijingganlu’ at 0.18 g/L, followed by ‘Beibinghong’, and the lowest level of lactic acid was in ‘Syrah’; the levels of acetic acid detected were all low, with the highest level of acetic acid in ‘Zuoshaner’, followed by ‘Beibinghong’ and ‘Zijingganlu’, and the lowest level in ‘Syrah’. The highest citric acid content was 0.89 g/L for ‘Zuoyouhong’, significantly higher than the other eight varieties, followed by ‘Syrah’, and the lowest citric acid content was 0.31 g/L for ‘Hassan’; the highest succinic acid content was 0.85 g/L for ‘Zuoshaner’, followed by ‘Beibinghong’ and ‘Zijingganlu’, and the lowest succinic acid content was 0.39 g/L for ‘Hassan’.

The cluster analysis results can better reflect the characteristics of the organic acid substances in the different wine samples (Figure 3). According to the cluster analysis of the organic acids of each variety, it can be seen that when the cross-cutting line takes values between 5 and 6, the nine varieties of wine samples can be divided into three categories: the first category is ’Beta‘ and ’Hassan’, the second category is ’Cabernet Sauvignon‘, and ’Syrah‘, and the third category is ‘Zuoyouhong‘, ‘Zijingganlu’, ‘Shuanghong’, ‘Zuoshaner’, and ‘Beibinghong’, indicating that the samples contained in each category have similarity in organic acids when the cross-cutting line takes values between 5 and 6. The result is also better in bringing together different types of grapes.

### 3.4. HS-GC-IMS Analysis of Wine Samples of Different Varieties

The aroma description of a wine is one of the keys to its quality [58]. The type and content of volatile compounds and their interactions are the main factors influencing the quality of grapes and wines and also determine the wine’s uniqueness [59]. Gas chromatography–mass spectrometry (HS-GC-MS) is a commonly used method for separating and quantifying volatile compounds in foodstuffs [60].

#### 3.4.1. Fingerprinting of Volatile Components of Wine Samples from Different Varieties

The fingerprints of the volatile flavor compounds of different wine varieties were constructed based on all the peaks in the HS-GC-IMS two-dimensional profile (Figure 4). Each sample was measured in parallel, with darker colors indicating higher peak intensities and higher contents. The fingerprints revealed the composition of and differences in the volatile flavor compounds of the wine samples of different varieties. As the graph shows, ‘Hassan’ has a high content of ethyl acetate, 1-propanol, pentanal, 2-pentanone, and 4-methyl-2-pentanone; ‘Zuoshaner’ has a high content of 1-penten-3-ol, 2,5-dimethylfuran, ethyl isobutyrate, and methyl acetate. The content of substances such as 1-butanol is higher in ‘Zuoyouhong’; 1-pentanol, 1-hydroxy-2-propanone, 2-Methylpropanal, and isobutyl acetate are higher in Beta; acetone, 1-hexanol, 2-butanol, Ethyl hexanoate, hexyl acetate, Ethyl octanoate, isoamyl acetate, Ethyl butanoate, Ethyl propanoate, and isobutyl butyrate, in ‘Zijingganlu’. Acetaldehyde is present in higher amounts; ethyl lactate and acetic acid are present in higher amounts in ‘Syrah’.

#### 3.4.2. Two-Dimensional Mapping of Wine Samples of Different Varieties

Significant differences were observed in the fingerprints of the nine wine samples (Figure 5). The differences were mainly in the content, with the color representing the concentration of the substance, white indicating a lower concentration, red a higher concentration, and darker colors indicating a greater concentration. HS-GC-IMS well separated the volatile substances in the nine wine samples, and the differences can be visualized.

Using the Hassan variety of wine as a reference, the rest of the spectrum was subtracted from the signal peaks in the Hassan to obtain a spectrum of the difference between the two (Figure 6). The blue area indicates that the substance is lower in this sample than in the Hassan wine, while the red area indicates that the substance is more present in this sample than in the Hassan wine. Again, the darker the color, the more significant the difference. The difference spectrum shows that ethyl acetate, 1-propanol, pentanal, 2-pentanone, and 4-methyl-2-pentanone are present in higher amounts in ‘Hassan’ than in the other varieties of wine.

### 3.5. Analysis of Volatile Matter Components

Aroma is one of the essential sensory characteristics of wine. A total of 52 typical volatile compounds were detected by qualitative analysis of the volatile components in the wine samples using the NIST database built into HS-GC-IMS and the IMS database (Table 6). The retention index is the calculation using N-ketones and views qualitative and quantitative analytical spectra and data with VO-Cal. with the most significant number of species detected being 20 esters, followed by 16 alcohols, 8 aldehydes, 4 ketones, 1 terpene, and 1 furan. The nine wine varieties had the same types of volatile aroma compounds detected, but the levels varied significantly. Of the nine wine varieties, ‘Hassan’ had the highest total volatile compound content of 58,160.24 µg/L, followed by ‘Beta’ 53,287.78 µg/L, ‘Syrah’ 52,144.75 µg/L, ‘Zijingganlu’ 43,863.56 µg/L, ‘Cabernet Sauvignon’ 43,678.17 µg/L, ‘Zuoshaner’ 42,201.41 µg/L, ‘Shuanghong’ 39,846.59 µg/L, ‘Zuoyouhong’ 37,567.72 µg/L, and ‘Beibinghong’ 34,187.72 µg/L. In terms of the proportion of each type of compound in the total volatile compounds of each variety, alcohols accounted for the most significant proportion of 58.34–62.21%, followed by esters with 28.29–32.21%. Alcohols and esters were the main aroma compounds in the nine wine samples.

#### 3.5.1. Esters

Esters are the most abundant compounds detected in each variety. Some important esters, such as ethyl butyrate, isoamyl acetate, and ethyl caproate, contribute to the desirable fruit organoleptic characteristics of the wine, including fruity aromas such as banana, strawberry, and green apple [65,66,67]. Among the esters detected were ethyl isovalerate, which contributes a fruity aroma with wine notes, isoamyl acetate with a banana odor, ethyl acetate with a sweet fruit flavor, and isobutyl propionate with a rum odor, and ethyl lactate with a pungent odor, etc. The aroma descriptions of the esters show that the esters are mainly fruit flavors, with ethyl acetate best reflecting the fruit aroma. Among the nine samples, ‘Hassan’ had the highest ethyl acetate content of 10,418.17 µg/L, followed by ‘Beta’ 9466.89 µg/L, ‘Syrah’ 9151.87 µg/L, ‘Cabernet Sauvignon’ 8984.57 µg/L, ‘Zijingganlu’ 8820.4 µg/L, ‘Shuanghong’ 8695.02 µg/L, ‘Zuoyouhong’ 8666.27 µg/L, ‘Beibinghong’ 8580.84 µg/L, and ‘Zuoshaner’ 8558.08 µg/L.

#### 3.5.2. Alcohols

Alcoholic substances make up the most significant proportion of the varieties, and their aroma is mainly reflected in grassy and alcoholic notes. The highest alcohol content was in ‘Zuoshaner’ 42,771.16 µg/L, followed by ‘Beibinghong’ 42,685.26 µg/L, ‘Zuoyouhong’ 41,757.22 µg/L, ‘Beta’ 41,379.82 µg/L, ‘Shuanghong’ 41,633.79 µg/L, ‘Zijingganlu’ 41,408.12 µg/L, ‘Hassan’ 40,590.22 µg/L, ‘Cabernet Sauvignon’ 39,997.59 µg/L, and ‘Syrah’ 39,858.69 µg/L.

#### 3.5.3. Aldehydes

The aldehyde content of each species ranged from 3.23% to 3.75%, with the highest content being 2645.91 µg/L for ‘Beta’. Followed by ‘Zijingganlu’ 2593.63 µg/L, ‘Hassan’ 2436.12 µg/L, ‘Zuoshaner’ 2354.01 µg/L, ‘Zuoyouhong’ 2310.9 µg/L, ‘Shuanghong’ 2295.74 µg/L, ‘Beibinghong’ 2263.64 µg/L, ‘Syrah’ 2189.4 µg/L, and ‘Cabernet Sauvignon’ 2112.78 µg/L.

#### 3.5.4. Others

The total concentrations of ketones, acids, terpenoids, and furans were low, accounting for only 1.38–2.57%, 3.04–4.24%, 0.03–0.07%, and 0–0.1% of the respective samples of each species.

### 3.6. Principal Component Analysis (PCA) of Wine Samples

In order to better present and distinguish the differences between the different varieties of wine samples, the volatile compounds identified by HS-GC-IMS were analyzed by PCA. The nine samples were well differentiated according to their aroma characteristics and varietal. The unsupervised multidimensional statistical analysis method (PCA) was applied to the samples to discriminate the magnitude of variability between groups of samples, between subgroups, and between samples within groups for the different wines. The contribution of PC1 was 46.8%, and that of PC2 was 20.7%, and the nine groups of samples showed a clear trend of separation on the two-dimensional plot, with no outlier samples and good clustering of samples of the same wine type. The PCA results reflected a significant overall difference in aroma matter between the nine groups of samples and differentiated them from each other. As shown in Figure 7, the ‘Zuoshaner’, ‘Beibinghong’, ‘Beta’, ‘Shuanghong’, and ‘Zijingganlu’ samples were closer to each other, while the ‘Hassan’, ‘Zuoyouhong’, ‘Cabernet Sauvignon’, and ‘Syrah’ samples were farther apart, indicating a significant difference between the aromatic characteristics of the different samples.

### 3.7. Analysis of the Key Aroma Compounds OAV in Wine Samples of Different Varieties

It is generally accepted that components with an OAV greater than one may directly influence the overall flavor. Based on the qualitative and quantitative results of GC-IMS, the literature was used to find the threshold values of the corresponding aroma compounds in water and to calculate their OAV values [68,69,70,71,72]. It was calculated that a total of 14 aroma compounds with OAV greater than one were detected in nine wine samples (Table 7), and the study showed that the OAV values were proportional to the contribution of aroma. The highest number of aroma compounds with OAV > 1 was found in the lipid group, with 10 species: Ethyl hexanoate-M, Isoamyl acetate-M, Ethyl 3-methylbutanoate-M, Ethyl 3-methylbutanoate-D, Isoamyl acetate-D, Ethyl butanoate-D, Ethyl isobutyrate, Ethyl Acetate, Ethyl hexanoate-D, and Ethyl octanoate; the three aldehydes are 2-Methylpropanal, Acetaldehyde, and 2-Methylbutanal; and one furan is 2,5-dimethylfuran. The key compounds’ OAV values in the nine wine samples varied. However, in general, the esters had higher OAV values than the other compounds, with isoamyl acetate-M among the esters having the most significant OAV values of 55.07–75.62 and contributing more to the overall aroma. The predominance of fruit flavors in the esters likewise indicates that esters are one of the most crucial compound groups contributing to the overall aroma and that fruit flavors are an essential aromatic feature in wine aromas.

Heat Map Analysis, PCA Analysis, and Correlation Analysis in Aromatic Compounds with OAV > 1 in Nine Grapefruits of Compounds with OAV > 1 in Fruits of Different Varieties.

Hierarchical analysis was used to cluster the concentrations of volatile aroma compounds with OAV values greater than one in the nine sample wines, as seen from the heat map analysis of each fruit sample (Figure 8), with red indicating high expression of that aroma compound component in the sample and blue indicating low expression of that aroma compound in the sample. The concentrations of volatile aroma compounds with OAV values greater than one varied considerably among the various samples.

Principal component analysis (PCA) is a multivariate statistical analysis technique. By identifying several principal component factors to represent the many complex and hard-to-find variables in the original sample, regularities and differences between samples are then assessed based on the contribution of the principal component factors in different samples [73]. The PCA results clearly show (Figure 9) that, in a relatively independent space, PCA analysis of the concentrations of volatile aroma compounds with OAV values greater than one in nine wine samples extracted a total of two principal components, with cumulative contributions of up to 68.6% for PC1 and PC2. Among the different varietal wines, ‘Zijingganlu’ was in the first quadrant of the score, with positive values on both PC1 and PC2. The ‘Shuanghong’, ‘Zuoyouhong’, and ‘Zuoshaner’ samples were located in the second quadrant, with positive scores on PC1 and negative scores on PC2. ‘Beibinghong’ is located at the junction of quadrants one and two and is positive on PC1 and negative on PC2. ‘Beta’ is located at the junction of quadrants two and three and is negative on PC2. ‘Hassan’ is in the third quadrant and is negative on PC1 and PC2. The ‘Cabernet Sauvignon’ and ‘Syrah’ samples were in quadrant four, with negative numbers on PC1 and positive numbers on PC2. This indicates that compounds with OAV values greater than one for volatile aromatic content vary considerably between wine samples of different varieties.

High correlations between substances are indicated by a correlation coefficient between 0.8 and 1.0, strong correlations are indicated by a correlation coefficient between 0.6 and 0.8, moderate correlations are indicated by a correlation coefficient between 0.4 and 0.6, and weak correlations are indicated by a correlation coefficient between 0.2 and 0.4. Correlation coefficients between 0 and 0.2 indicate no or very weak correlation between substances at all.

In the correlation analysis of Figure 10, the red box’s Pearson correlation coefficient was significantly correlated. 2-Methylpropanal was strongly correlated with Isoamyl acetate-M, Ethyl 3-methylbutanoate-M, Ethyl 3-methylbutanoate-D, Ethyl butanoate-D, Ethyl isobutyrate, and Ethyl octanoate; Acetaldehyde is strongly correlated with Ethyl Acetate and Isoamyl acetate-D; 2-Methylbutanal is strongly correlated with Ethyl butanoate-D; Ethyl hexanoate-M is strongly correlated with 2,5 Dimethylfuran, Ethyl hexanoate-D, and Isoamyl acetate-D; Isoamyl acetate-M is strongly correlated with Ethyl isobutyrate, Ethyl butanoate-D, Ethyl 3-methylbutanoate-M and Ethyl 3-methylbutanoate-D; Ethyl 3-methylbutanoate-M is strongly correlated with Ethyl isobutyrate, Ethyl butanoate-D, and Ethyl 3-methylbutanoate-D; Ethyl 3-methylbutanoate-D is strongly correlated with Ethyl isobutyrate; Ethyl butanoate-D is strongly correlated with Ethyl octanoate and Ethyl isobutyrate; Ethyl hexanoate-D is strongly correlated with D-2,5 Dimethylfuran.

The Pearson correlation coefficients for the green boxes are significantly negatively correlated. 2-methylpropanal was significantly negatively correlated with Ethyl Acetate, Ethyl hexanoate-m was significantly negatively correlated with Acetaldehyde and Isoamyl acetate M, and 2-methylbutanal was significantly negatively correlated with Isoamyl acetate D. Isoamyl acetate-M was significantly negatively correlated with Ethyl acetate, Ethyl 3-methylbutanoate-M was significantly negatively correlated with Ethyl acetate. Ethyl 3-methylbutanoate-D was significantly negatively correlated with Ethyl Acetate, and Ethyl isobutyrate was significantly negatively correlated with Ethyl Acetate.

### 3.8. Analysis of Volatile Wine Compounds 0PLS-DA

OPLS-DA is a supervised statistical method for discriminant analysis that enables the identification of sample differences and the acquisition of characteristic markers of sample differences [74,75]. The contribution of each variable to wine flavor was further quantified based on the variable importance for the projection (VIP) in the OPLS-DA model, and volatile flavor compounds with VIP > 1 were screened as potential characteristic volatile markers [76]. In general, variables with VIP values > 1 are considered metabolites that cause differences between groups. Most studies use the content of volatile compounds as an evaluation indicator for OPLS-DA analysis, but some volatile compounds, although high in content, also have high threshold values that are not easily perceived by the human sense of smell, so in this experiment, the OAV values of compounds with OAV values > 1 in the composition of wine samples from different varieties were used as Y variables for OPLS-DA analysis (Figure 11), which can more accurately screen out differential volatile compounds. Wine aroma quality depends on the combined effect of several aroma compounds, and compounds with VIP values > 1 were screened. The results revealed that the compounds influencing aromatic differences might be related to ethyl isobutyrate, ethyl hexanoate-D, 2-methylpropanal, ethyl octanoate, ethyl butanoate-D, and Isoamyl acetate-D (Table 8).

### 3.9. Sensory Evaluation Characteristics of the Original Wine

The sensory evaluation of the quality of the nine wine samples was carried out using six descriptors: ‘Floral’, ‘Fruity’, ‘Plant and Herb’, ‘Fermented’, ‘Tarry’, and ‘Candy’. Statistical analysis showed that the samples differed in each descriptor (Figure 12). These significant differences suggest that the flavor intensity of each sample was significantly different. Although panelists were trained before sensory assessment, they may have significantly influenced the descriptor rating results. This phenomenon is not uncommon in characterization analyses. It suggests that panelists applied different levels of qualitative scoring due to physiological differences in perceived intensity or personal preferences (e.g., central or extreme raters). However, no significant interactions between panelists and replication were found in the study, suggesting that all panelists were duplicated in triplicate for all descriptors scored. Similarly, there were no significant interactions between sample and replication and panel members, suggesting that the sensory data were valid and credible. It is also clear from the GC-IMS results that, apart from alcohols, esters make up the most significant proportion of the total volatile compounds detected. It is common sense that esters contribute to the desirable fruit organoleptic characteristics of the wine, with the ‘fruity’ and ‘floral’ descriptors being the primary and most fundamental parts of the global flavor of the wine. These two descriptors are therefore important indicators of the quality of a wine’s aromas. Compared with other samples, ‘Beta’ showed a higher ‘floral’ and ‘fruity’ aroma, which was the same as the GC-IMS results. Among the nine wine varieties, the content of ester compounds in ‘Beta’ accounted for the highest proportion of the total content of aroma compounds. ‘Shuanghong’ shows high levels of vegetal and herbal aromas, with hexanal, valeraldehyde, (E)-2-heptanal, and (E)-2-hexenal probably closely related to the vegetal and herbal descriptors, i.e., aldehydes and alcohols are often associated with ‘green’, ‘fresh grass’, and ‘green plants. A total of eight aldehydes were detected in the GC-IMS results, with acetaldehyde being the main aldehyde, ‘Shuanghong’ having a high acetaldehyde content of 1941.76 μg-L-1 among the nine varietal wines. The fermentation aromas were mainly produced by the fermentation and aging stages, with all varieties scoring low in tar aromas. The tar aromas were mainly produced by changes in tannins during fermentation, ‘Zijingganlu’ showing a higher tar and fermentation aroma. The sensory evaluation results also showed significant differences in the fruit aromas of the nine varieties, indicating that the results of the sensory evaluation were similar to the results of the data analysis. Combined with the aroma characteristics of several aroma compounds with VIP values greater than one calculated from the OPLS-DA analysis, it was found that the fruit aromas were the main compounds in the wine aroma substances, also indicating that the six compounds with VIP values greater than one calculated based on the OAV values were compounds that could influence the differences between groups, similar to the results of the sensory evaluation.

## 4. Conclusions

In this study, nine grape samples from the National fruit tree germplasm *Vitis amurensis* nursery in Zuojia town were collected and used to produce the original grape wine. The wines’ basic physical and chemical properties, color, and aromatic composition were examined, and the sensory evaluation of the nine varieties was carried out. The test results showed that ‘Hassan’ had the highest solids content, ‘Zuoshaner’ had the highest total acid, residual sugar, total anthocyanin, and total phenol content, and ‘Shuanghong’ had the highest tannin content. ‘Cabernet Sauvignon’ had the brightest color of the nine varieties, and ‘Zuoshaner’ had the most pronounced red and yellow hues, with the most excellent saturation. The HS-GC-IMS technique was used to analyze the variation in volatile flavor compounds of raw wine samples of different varieties. A total of 52 volatile flavor substances were identified, including 20 esters, 16 alcohols, 8 aldehydes, 4 ketones, and 1 terpene and furan each, with significant differences in the content of volatile flavor substances in different varieties of wine. The specific wine aroma characteristics were characterized based on the volatile compounds and by quantitative descriptive analysis of the data through multivariate statistical analysis. In contrast, the key volatile compounds affecting the wine aroma were screened by combining principal component analysis, OAV value analysis, and VIP value analysis. Fourteen volatile aroma compounds with OAV values greater than one were screened. In general, the OAV values of esters were higher than those of other compounds and contributed more to the overall aroma. The OAV values of compounds with OAV values greater than one in the composition of wine samples from different varieties were used as Y variables for OPLS-DA analysis to obtain characteristic markers of sample variation. The results revealed that compounds influencing aroma variation might be related to ethyl isobutyrate, ethyl hexanoate-D, 2-methylpropanal, ethyl octanoate, ethyl butanoate-D, and Isoamyl acetate-D. Through QDA analysis of the aroma components of the nine varieties, the sensory evaluation results were consistent with the analysis results. The main factors affecting the differences in wine flavor are variety, cultivation techniques, environment, cultivation techniques, environment, fermentation process, aging time and storage conditions, etc. This study reflects the differences between different varieties of wines to a certain extent through micro-wine making and provides a reference for wine development and promotion. In the future, the flavor of wine can be enhanced from different angles to develop wines with Chinese characteristics and increase the share of Chinese wines on the international market.

## Figures and Tables

**Figure 1 foods-11-02767-f001:**
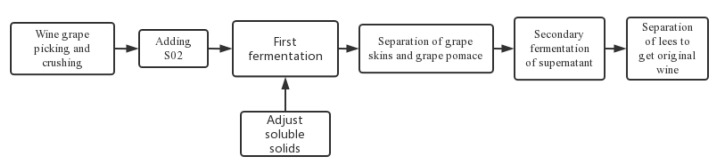
Fermentation flow chart for wine.

**Figure 2 foods-11-02767-f002:**
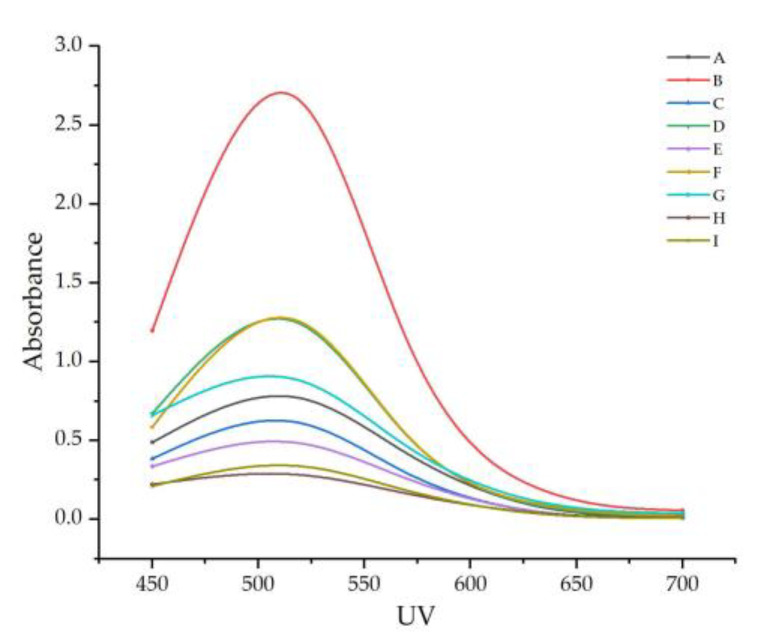
The change rule of the absorbance in the visible band of the sample. Note: From A to I are ‘Hassan’, ‘Zuoshaner’, ‘Beibinghong’, ‘Zuoyouhong’, ‘Beta’, ‘Shuanghong’, ‘Zijingganlu’, ‘Cabernet Sauvignon’, and ‘Syrah’.

**Figure 3 foods-11-02767-f003:**
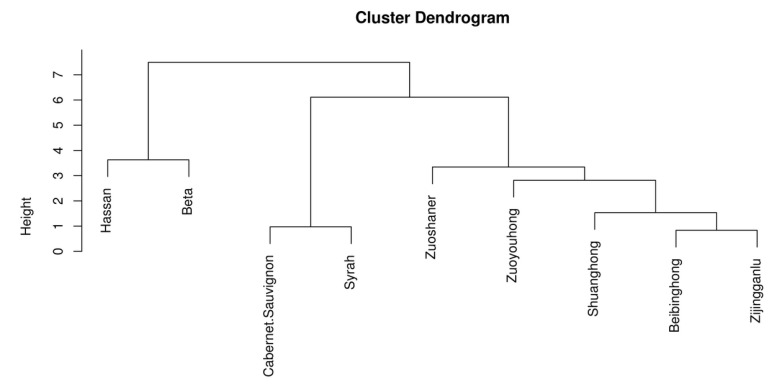
Cluster analysis chart of organic acid content in different wine samples.

**Figure 4 foods-11-02767-f004:**
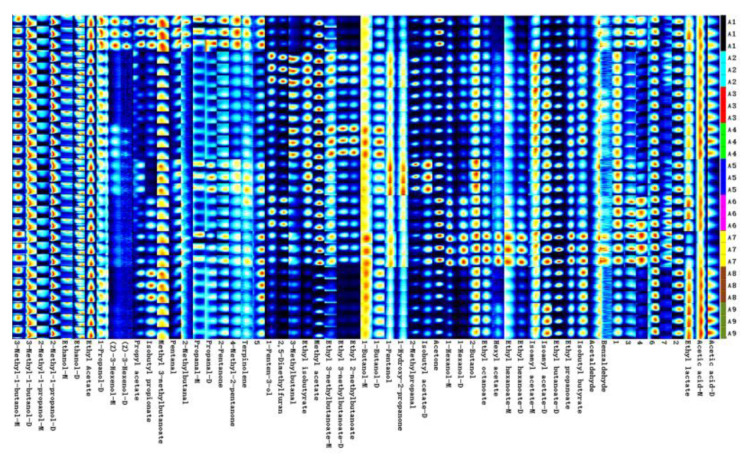
Fingerprints of volatile compounds in different wine varieties. Note: From A1 to A9 are ‘Hassan’, ‘Zuoshaner’, ‘Beibinghong’, ‘Zuoyouhong’, ‘Beta’, ‘Shuanghong’, ‘Zijingganlu’, ‘Cabernet Sauvignon’, and ‘Syrah’.

**Figure 5 foods-11-02767-f005:**
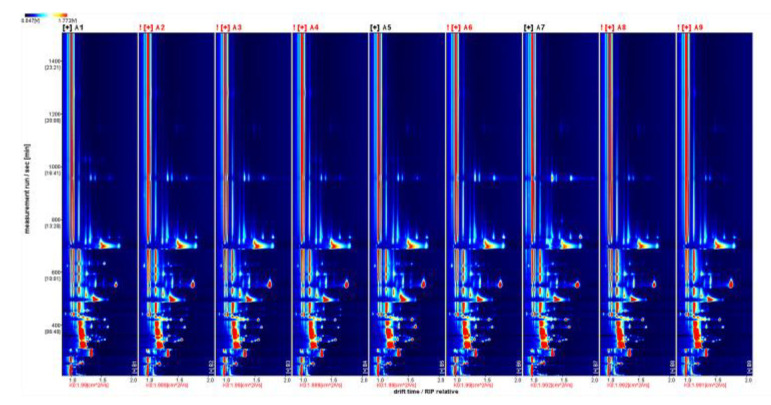
HS-GC-IMS 2D spectrum (top view). Note: From A1 to A9 are ‘Hassan’, ‘Zuoshaner’, ‘Beibinghong’, ‘Zuoyouhong’, ‘Beta’, ‘Shuanghong’, ‘Zijingganlu’, ‘Cabernet Sauvignon’, and ‘Syrah’. The entire graph has a blue background, and the red vertical line at the horizontal coordinate 1.0 is the RIP peak (reactive ion peak, normalized). Each point on either side of the RIP peak represents a volatile organic compound.

**Figure 6 foods-11-02767-f006:**
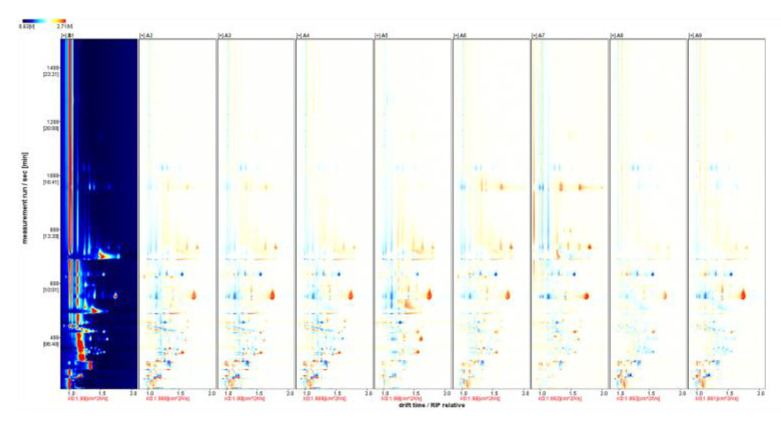
HS-GC-IMS difference spectrum of the sample. Note: From A1 to A9 are ‘Hassan’, ‘Zuoshaner’, ‘Beibinghong’, ‘Zuoyouhong’, ‘Beta’, ‘Shuanghong’, ‘Zijingganlu’, ‘Cabernet Sauvignon’, ‘Syrah’. Note: From A1 to A9 are ‘Hassan’, ‘Zuoshaner’, ‘Beibinghong’, ‘Zuoyouhong’, ‘Beta’, ‘Shuanghong’, ‘Zijingganlu’, ‘Cabernet Sauvignon’, and ‘Syrah’.

**Figure 7 foods-11-02767-f007:**
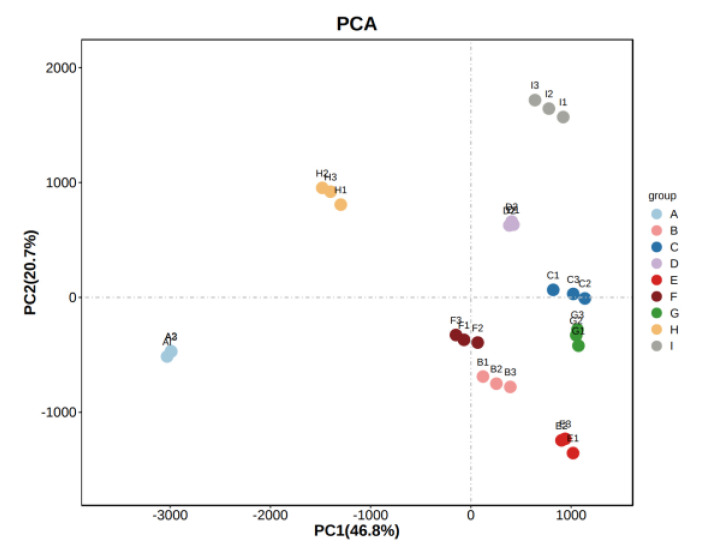
PCA analysis of the sample. Note: From A to I are ‘Hassan’, ‘Zuoshaner’, ‘Beibinghong’, ‘Zuoyouhong’, ‘Beta’, ‘Shuanghong’, ‘Zijingganlu’, ‘Cabernet Sauvignon’, and ‘Syrah’.

**Figure 8 foods-11-02767-f008:**
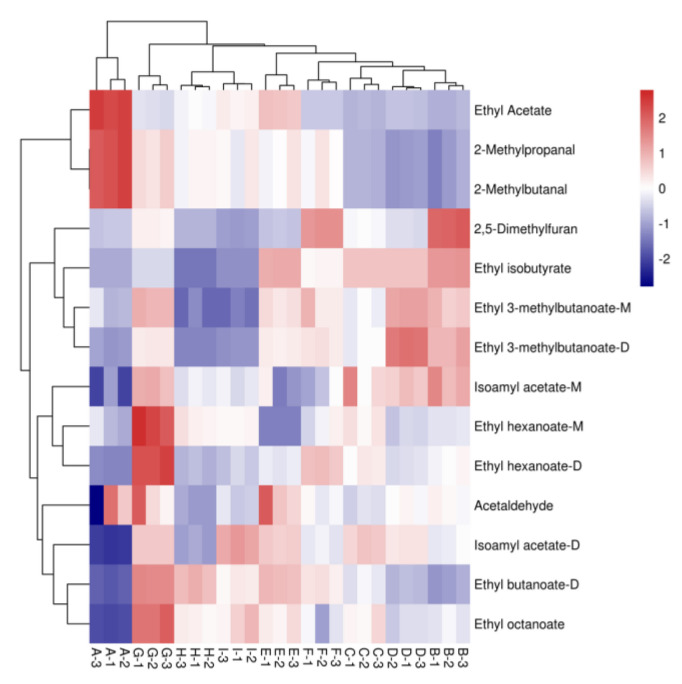
Heat map analysis of aromatic compounds with OAV > 1 in nine varietal wine samples. Note: From A to I are ‘Hassan’, ‘Zuoshaner’, ‘Beibinghong’, ‘Zuoyouhong’, ‘Beta’, ‘Shuanghong’, ‘Zijingganlu’, ‘Cabernet Sauvignon’, ‘Syrah’. Note: From A to I are ‘Hassan’, ‘Zuoshaner’, ‘Beibinghong’, ‘Zuoyouhong’, ‘Beta’, ‘Shuanghong’, ‘Zijingganlu’, ‘Cabernet Sauvignon’, and ‘Syrah’.

**Figure 9 foods-11-02767-f009:**
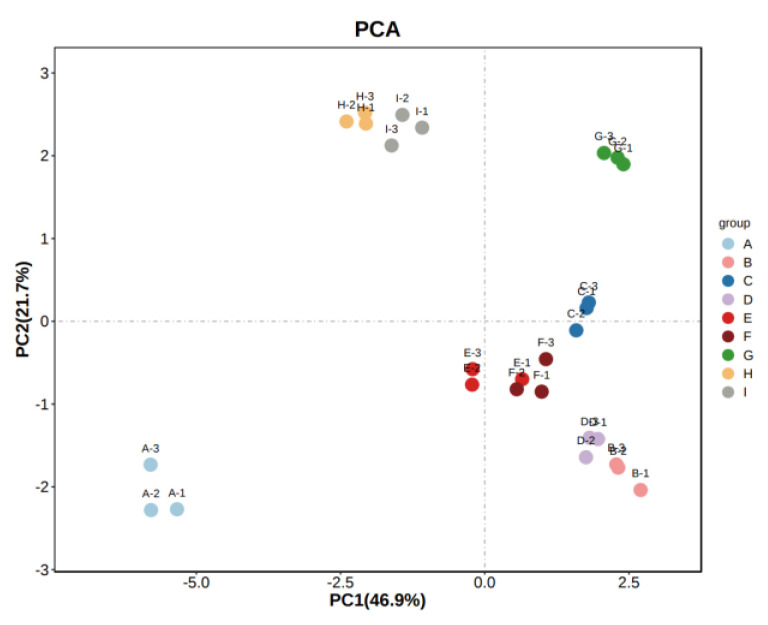
Loading diagram of PCA analysis of volatile aroma compounds with OAV values greater than 1 in nine wine varieties.

**Figure 10 foods-11-02767-f010:**
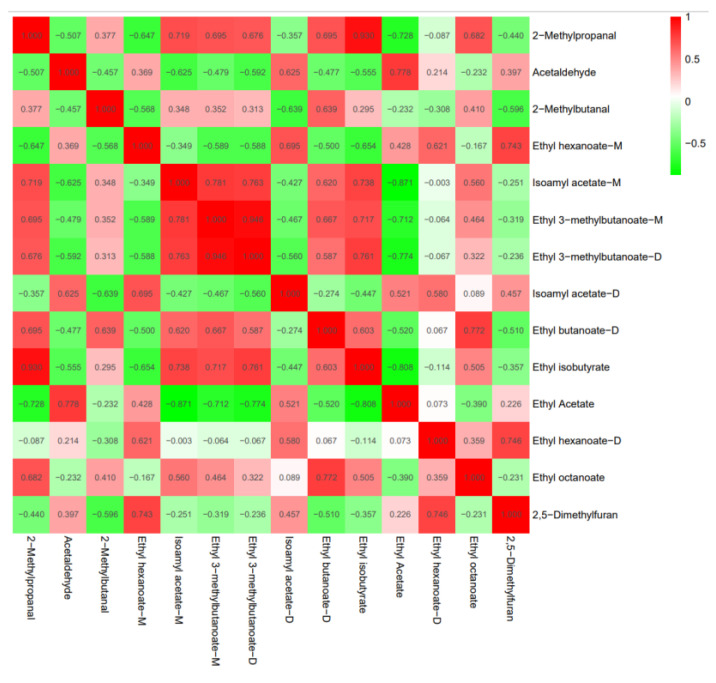
Analyses of the correlation between aroma compounds and OAV greater than 1 in nine grape wine varieties.

**Figure 11 foods-11-02767-f011:**
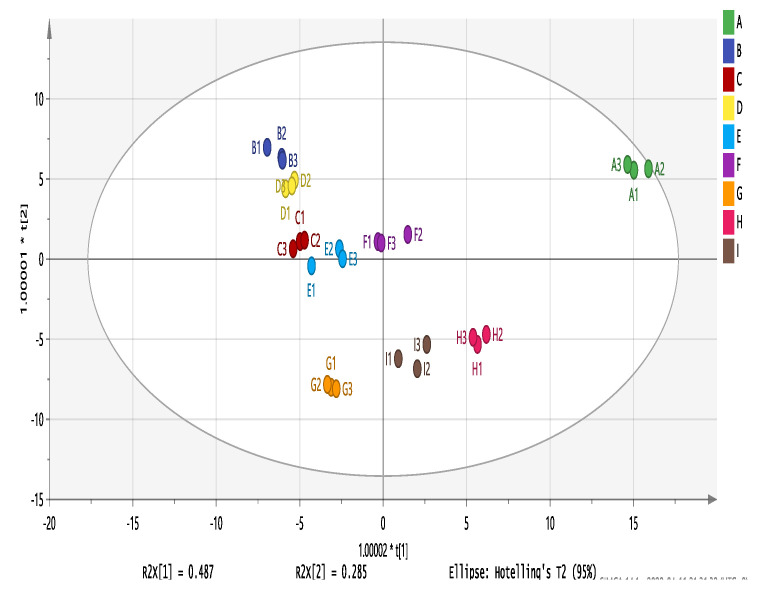
OPLS-DA analysis of nine wine varieties. Note: From A to I are ‘Hassan’, ‘Zuoshaner’, ‘Beibinghong’, ‘Zuoyouhong’, ‘Beta’, ‘Shuanghong’, ‘Zijingganlu’, ‘Cabernet Sauvignon’, and ‘Syrah’.

**Figure 12 foods-11-02767-f012:**
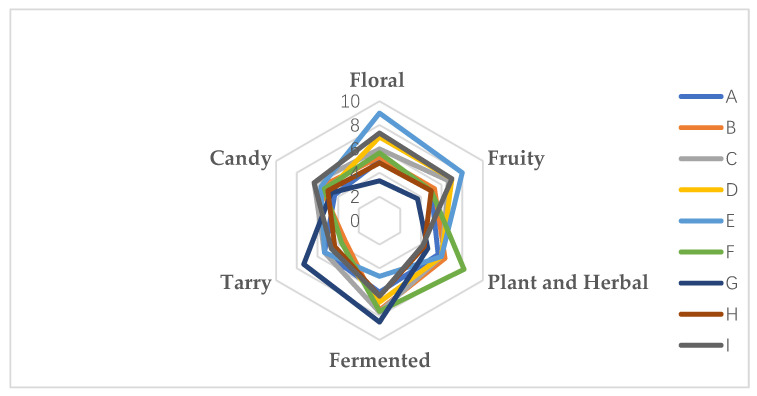
Description of the quantitative aroma analysis of different wine varieties. Note: From A to I are ‘Hassan’, ‘Zuoshaner’, ‘Beibinghong’, ‘Zuoyouhong’, ‘Beta’, ‘Shuanghong’, ‘Zijingganlu’, ‘Cabernet Sauvignon’, and ‘Syrah’.

**Table 1 foods-11-02767-t001:** Organic acid standard curve.

Name	Standard Curves	R2
Tartaric acid	f(x) = 951.0962x + 39.1049	0.9954
Malic acid	f(x) = 524.5688x + 28.0833	0.9927
Lactic acid	f(x) = 5993.8264x − 2.4190	0.9999
Iceacetic acid	f(x) = 8221.6410x + 13.4295	0.9979
Citric acid	f(x) = 1492.4554x + 3.8029	0.9994
Succinic acid	f(x) = 873.4799x − 22.9056	0.9989

**Table 2 foods-11-02767-t002:** Gas chromatography conditions.

Time (Min: Sec)	E1 (Drift Gas)	E2 (Carrier Gas)	Recording
00:00,000	150 mL/min	2 mL/min	rec
02:00,000	150 mL/min	2 mL/min	-
10:00,000	150 mL/min	10 mL/min	-
20:00,000	150 mL/min	100 mL/min	-
30:00,000	150 mL/min	100 mL/min	stop

**Table 3 foods-11-02767-t003:** The basic physical and chemical indicators of the original wine.

Varieties	Solids (°Brix)	Titratable Acid (g/L)	Total Sugar (g/L)	Total Anthocyanin (mg/L)	Total Phenols (g/L)	Tannin (g/L)	pH Value	Dry Extract/g/L	Alcohol by Volume (*v*/*v*)
‘Hassan’	9.40 ± 0.00 ^a^	11.00 ± 0.29 ^g^	4.10 ± 0.12 ^c^	151.96 ± 6.68 ^f^	1.61 ± 0.04 ^e^	2.33 ± 0.31 ^d^	4.13 ± 0.01 ^b^	31.50 ± 0.00 ^d^	13.00 ± 0.00 ^a^
‘Zuoshaner’	9.17 ± 0.06 ^b^	16.25 ± 0.29 ^a^	5.70 ± 0.35 ^a^	1477.85 ± 48.31 ^a^	3.56 ± 0.04 ^a^	3.59 ± 0.07 ^b^	3.65 ± 0.01 ^h^	40.80 ± 0.17 ^a^	11.00 ± 0.00 ^c^
‘Beibinghong’	8.53 ± 0.06 ^e^	12.69 ± 0.43 ^e^	2.90 ± 0.24 ^f^	236.57 ± 14.97 ^e^	1.39 ± 0.03 ^f^	1.76 ± 0.05 ^e^	3.79 ± 0.01 ^g^	30.90 ± 0.17 ^e^	12.00 ± 0.00 ^b^
‘Zuoyouhong’	8.77 ± 0.06 ^c^	13.63 ± 0.29 ^c^	3.60 ± 0.17 ^d^	600.60 ± 33.83 ^c^	2.17 ± 0.05 ^c^	2.83 ± 0.21 ^c^	3.65 ± 0.00 ^h^	33.23 ± 0.12 ^c^	12.00 ± 0.00 ^b^
‘Beta’	8.7 ± 0.10 ^d^	15.5 ± 0.39 ^b^	3.13 ± 0.23 ^e^	95.18 ± 9.30 ^h^	1.14 ± 0.04 ^g^	1.50 ± 0.13 ^g^	3.90 ± 0.01 ^e^	33.10 ± 0.00 ^c^	12.00 ± 0.00 ^b^
‘Shuanghong’	8.67 ± 0.58 ^d^	12.75 ± 0.00 ^d^	4.46 ± 0.21 ^b^	1440.56 ± 75.15 ^b^	3.00 ± 0.24 ^b^	3.64 ± 0.09 ^a^	3.84 ± 0.01 ^f^	37.13 ± 0.12 ^b^	13.00 ± 0.00 ^a^
‘Zijingganlu’	7.9 ± 0.00 ^g^	7.38 ± 0.29 ^i^	4.10 ± 0.12 ^c^	496.51 ± 10.74 ^d^	1.90 ± 0.10 ^d^	1.67 ± 0.10 ^f^	4.12 ± 0.01 ^c^	37.40 ± 0.17 ^b^	12.00 ± 0.00 ^b^
‘Cabernet Sauvignon’	8.1 ± 0.00 ^f^	8.25 ± 0.19 ^h^	1.65 ± 0.07 ^h^	108.54 ± 7.28 ^g^	0.82 ± 0.08 ^h^	1.36 ± 0.34 ^h^	4.39 ± 0.01 ^a^	27.30 ± 0.17 ^f^	12.00 ± 0.00 ^b^
‘Syrah’	8.83 ± 0.06 ^c^	11.31 ± 0.39 ^f^	2.47 ± 0.17 ^g^	83.49 ± 1.67 ^i^	0.70 ± 0.03 ^i^	1.30 ± 0.31 ^i^	4.05 ± 0.01 ^d^	18.70 ± 0.17 ^g^	13.00 ± 0.00 ^a^

Means with different letters in the same column express significant differences (Duncan’s test *p* < 0.05).

**Table 4 foods-11-02767-t004:** CIELab parameters of different original wines.

Varieties	L*	a*	b*	Cab*	hab*	ΔEab*
‘Hassan’	66.91 ± 0.06 ^e^	40.15 ± 0.01 ^d^	7 ± 0.28 ^e^	40.75 ± 0.21 ^e^	9.89 ± 0.48 ^f^	29.16 ^e^
‘Zuoshaner’	41.65 ± 0.10 ^i^	167.48 ± 0.07 ^a^	21.76 ± 0.17 ^a^	168.89 ± 0.10 ^a^	7.4 ± 0.77 ^g^	157.30 ^a^
‘Beibinghong’	73.76 ± 0.12 ^d^	35.46 ± 0.07 ^f^	7.24 ± 0.30 ^d^	36.2 ± 0.14 ^f^	11.54 ± 0.28 ^e^	21.46 ^f^
‘Zuoyouhong’	59.72 ± 0.04 ^g^	52 ± 0.1 ^c^	13.12 ± 0.11 ^c^	53.63 ± 0.69 ^b^	14.16 ± 0.54 ^c^	43.59 ^c^
‘Beta’	76.97 ± 0.06 ^c^	28.87 ± 0.14 ^g^	6.87 ± 0.24 ^f^	29.68 ± 0.14 ^g^	13.39 ± 0.17 ^d^	14.15 ^g^
‘Shuanghong’	59.65 ± 0.11 ^h^	53.3 ± 0.26 ^b^	4.95 ± 0.07 ^h^	53.53 ± 0.24 ^c^	5.31 ± 0.27 ^h^	44.04 ^b^
‘Zijingganlu’	63.22 ± 0.11 ^f^	39.84 ± 0.07 ^e^	17.34 ± 0.28 ^b^	43.46 ± 0.19 ^d^	23.52 ± 0.4 ^a^	33.43 ^d^
‘Cabernet Sauvignon’	83.01 ± 0.06 ^b^	22.48 ± 0.06 ^h^	1.58 ± 0.19 ^i^	22.54 ± 0.07 ^h^	4.02 ± 0.07 ^i^	6.94 ^h^
‘Syrah’	84.65 ± 0.12 ^a^	17.05 ± 0.14 ^i^	5.57 ± 0.11 ^g^	17.93 ± 0.11 ^i^	18.09 ± 0.25 ^b^	

Means with different letters in the same column express significant differences (Duncan’s test *p* < 0.05).

**Table 5 foods-11-02767-t005:** Comparison of Organic Acids in Different Variety Wine Samples.

Varieties	Tartaric Acid/g/L	Malic Acid/g/L	Lactic Acid/g/L	Acetic Acid/g/L	Citric Acid/g/L	Succinic Acid/g/L
‘Hassan’	3.5 ± 0.14 ^f^	5.97 ± 0.2 ^c^	0.18 ± 0 ^a^	0.09 ± 0 ^e^	0.31 ± 0.06 ^i^	0.39 ± 0.04 ^i^
‘Zuoshaner’	7.99 ± 0.02 ^a^	6.26 ± 0.22 ^b^	0.11 ± 0.02 ^d^	0.13 ± 0.02 ^a^	0.45 ± 0.02 ^f^	0.85 ± 0.05 ^a^
‘Beibinghong’	6.13 ± 0.1 ^c^	5.17 ± 0.16 ^e^	0.13 ± 0.01 ^b^	0.12 ± 0 ^b^	0.42 ± 0.04 ^g^	0.83 ± 0.14 ^b^
‘Zuoyouhong’	6.84 ± 0.03 ^b^	3.28 ± 0.08 ^g^	0.1 ± 0 ^e^	0.09 ± 0 ^e^	0.89 ± 0.43 ^a^	0.64 ± 0.1 ^f^
‘Beta’	3.2 ± 0.51 ^g^	9.51 ± 0.04 ^a^	0.11 ± 0.02 ^d^	0.1 ± 0.01 ^d^	0.41 ± 0.02 ^h^	0.58 ± 0.02 ^g^
‘Shuanghong’	4.82 ± 0.16 ^d^	5.22 ± 0 ^d^	0.11 ± 0.01 ^d^	0.11 ± 0.01 ^c^	0.59 ± 0.07 ^c^	0.68 ± 0.02 ^e^
‘Zijingganlu’	6.08 ± 3.39 ^c^	4.35 ± 0.22 ^f^	0.18 ± 0 ^a^	0.12 ± 0 ^b^	0.54 ± 0.06 ^d^	0.75 ± 0.1 ^c^
‘Cabernet Sauvignon’	3.16 ± 0.34 ^g^	2.85 ± 0.21 ^h^	0.12 ± 0.02 ^c^	0.11 ± 0.02 ^c^	0.48 ± 0.02 ^e^	0.69 ± 0.02 ^d^
‘Syrah’	3.6 ± 0.29 ^e^	2.03 ± 2.2 ^i^	0.04 ± 0.04 ^f^	0.07 ± 0 ^f^	0.65 ± 0.1 ^b^	0.48 ± 0.29 ^h^

Means with different letters in the same column express significant differences (Duncan’s test *p* < 0.05).

**Table 6 foods-11-02767-t006:** Composition of volatile compounds in different varieties of wine grapes.

Serial Number	Aromatic Substances	Aroma Description [61,62,63,64]	Aroma Content of Dry Wine (μg·L^−1^)								
			‘Hassan’	‘Zuoshaner’	‘Beibinghong’	‘Zuoyouhong’	‘Beta’	‘Shuanghong’	‘Zijingganlu’	‘Cabernet Sauvignon’	‘Syrah’
1	(Z)-3-Hexenol-M	Green, herbal and green leafy aromas	219.93 ± 5.82 ^a^	83.6 ± 5.27 ^e^	75.31 ± 2.25 ^e, f^	146.84 ± 2.45 ^b^	123.7 ± 1.06 ^c^	71.48 ± 2.49 ^f^	95.83 ± 2.04 ^d^	56.8 ± 7.98 ^g^	67.5 ± 4.48 ^f^
2	(Z)-3-Hexenol-D	Green, herbal and green leafy aromas	111.83 ± 7.15 ^a^	50.28 ± 10.43 ^c^	45.79 ± 8.90 ^c^	68.03 ± 2.54 ^b^	57.05 ± 4.74 ^b, c^	53.94 ± 6.40 ^c^	49.75 ± 2.86 ^c^	44.18 ± 4.76 ^c^	48.35 ± 6.59 ^c^
3	1-Hexanol-M	Fruity, grassy notes, toast	993.2 ± 31.14 ^e^	1489.47 ± 33.34 ^c^	1458.09 ± 27.54 ^c^	1047.64 ± 10.79 ^e^	1176.86 ± 23.21 ^d^	1785.85 ± 39.79 ^b^	2148.75 ± 22.60 ^a^	866.76 ± 43.55 ^f^	763.5 ± 54.40 ^g^
4	1-Hexanol-D	Fruity, grassy notes, toast	250.77 ± 7.33 ^e^	531.65 ± 27.26 ^c^	520.52 ± 35.14 ^c, f^	255.52 ± 3.06 ^e^	334.53 ± 8.60 ^d^	852.4 ± 56.36 ^b^	1086.67 ± 44.75 ^a^	193.55 ± 22.34 ^f, g^	147.21 ± 8.75 ^g^
5	3-Methyl-1-butanol-M	Mellow, astringent	943.59 ± 43.05 ^a^	812.45 ± 20.57 ^cd^	851.48 ± 58.57 ^a, b, c^	832.52 ± 12.27 ^b, c^	728.86 ± 7.58 ^d^	788.4 ± 26.71 ^c, d^	935.34 ± 105.14 ^a^	922.77 ± 22.03 ^b^	851.84 ± 34.46 ^a, b, c^
6	3-Methyl-1-butanol-D	Mellow, astringent	8554.48 ± 65.80 ^b^	9002.29 ± 61.26 ^a^	8917.16 ± 29.07 ^a^	9007.77 ± 17.74 ^a^	8822.4 ± 32.81 ^a^	8833.33 ± 73.64 ^a^	7499.75 ± 226.82 ^c^	8501.83 ± 56.82 ^b^	8623.55 ± 98.46 ^b^
7	1-Penten-3-ol	Fruity	180.15 ± 6.58 ^g^	422.62 ± 3.88 ^a^	273.28 ± 2.76 ^c^	207.08 ± 3.72 ^f^	253.62 ± 7.64 ^d^	362.33 ± 8.57 ^b^	231.4 ± 6.16 ^e^	153.23 ± 2.16 ^h^	126.9 ± 1.08 ^i^
8	1-Butanol-M	Special odors	30.89 ± 2.21 ^e^	46.04 ± 0.63 ^b^	44.76 ± 2.24 ^b^	49.88 ± 0.43 ^a^	41.02 ± 1.30 ^c^	35.95 ± 1.11 ^d^	37.56 ± 1.29 ^d^	40.67 ± 0.65 ^c^	41.79 ± 0.70 ^c^
9	2-Methyl-1-propanol-M		599.96 ± 19.24 ^b, c^	485.74 ± 5.37 ^f^	520.61 ± 15.98 ^e^	569.07 ± 2.20 ^d^	422.76 ± 4.62 ^g^	592.53 ± 6.57 ^c^	614.99 ± 4.60 ^b^	672.12 ± 6.35 ^a^	688.27 ± 1.72 ^a^
10	Ethanol-M	wine and the pungent, spicy odor	1311.88 ± 4.81 ^d^	1412.74 ± 3.20 ^c^	1304.91 ± 41.78 ^d^	1249.25 ± 7.25 ^d^	1017.47 ± 42.28 ^e^	1400.59 ± 53.93 ^c^	1253.61 ± 13.68 ^d^	1569.31 ± 41.81 ^a^	1496.54 ± 39.95 ^b^
11	Ethanol-D	wine and the pungent, spicy odor	17,200.55 ± 166.71 ^e^	17,352.84 ± 34.20 ^d^	17,798.05 ± 26.27 ^b^	18,102.52 ± 22.43 ^a^	16,638.7 ± 46.46 ^g^	16,941.58 ± 25.13 ^f^	17,386.46 ± 12.28 ^d^	17,448.73 ± 89.54 ^d^	17,608.47 ± 82.23 ^c^
12	2-Butanol	Wine-like odor	593.84 ± 8.65 ^e^	652.72 ± 3.55 ^d^	680.19 ± 7.70 ^c^	536.98 ± 5.95 ^f^	825.02 ± 4.29 ^a^	672.46 ± 1.25 ^c^	734.18 ± 5.04 ^b^	650.18 ± 3.60 ^d^	642.3 ± 1.67 ^d^
13	1-Propanol-D	Alcohol, ripe fruit flavors	1927.28 ± 43.20 ^a^	1566.62 ± 8.78 ^c^	1707.77 ± 10.83 ^b^	1668.13 ± 6.05 ^b^	1355.04 ± 17.80 ^e^	1312.49 ± 31.82 ^ef^	1292.1 ± 25.09 ^f^	1430.85 ± 1.21 ^d^	1414.43 ± 26.39 ^d^
14	1-Butanol-D	Special odors	140.48 ± 2.50 ^f^	173.58 ± 2.91 ^cd^	161.83 ± 8.88 ^d^	251.07 ± 2.13 ^a^	186.02 ± 3.73 ^c^	155.53 ± 9.21 ^d^	172.18 ± 2.93 ^d^	211.73 ± 4.79 ^b^	202.27 ± 13.18 ^b^
15	2-Methyl-1-propanol-D	Alcoholic, Solvent odor, bitter	7330.75 ± 11.62 ^f^	8376.08 ± 72.80 ^b^	8048.6 ± 94.01 ^c^	7487.12 ± 14.88 ^e^	9058.24 ± 33.52 ^a^	7503.18 ± 64.89 ^e^	7637.66 ± 23.50 ^d^	7003.94 ± 79.59 ^g^	6898.94 ± 104.89 ^g^
16	1-Pentanol		200.64 ± 7.80 ^e^	312.44 ± 1.83 ^b^	276.91 ± 2.74 ^c^	277.8 ± 5.20 ^c^	338.53 ± 4.28 ^a^	271.75 ± 1.42 ^c^	231.89 ± 3.68 ^d^	230.94 ± 5.58 ^d^	236.83 ± 6.41 ^d^
**Alcohols**	**Subtotal**		40,590.22	42,771.16	42,685.26	41,757.22	41,379.82	41,633.79	41,408.12	39,997.59	39,858.69
	**Percentage**		61.38%	61.29%	60.91%	60.86%	58.63%	60.27%	58.34%	62.21%	59.58%
1	Acetone	Peculiar pungent odor	1343.69 ± 8.95 ^b^	796.44 ± 5.12 ^f^	690.35 ± 4.18 ^h^	734.68 ± 7.79 ^g^	961.93 ± 17.62 ^d^	1273.48 ± 6.76 ^c^	1400.14 ± 22.29 ^a^	862.79 ± 12.17 ^e^	756.29 ± 6.78 ^g^
2	4-Methyl-2-pentanone	Pleasant keto-like fragrance	108.68 ± 0.44 ^a^	44.75 ± 1.02 ^e^	44.49 ± 0.53 ^e^	38.13 ± 1.20 ^f^	92.75 ± 0.55 ^b^	56.24 ± 0.82 ^d^	59.14 ± 3.98 ^d^	62.94 ± 2.04 ^c^	63.08 ± 2.42 ^c^
3	2-Pentanone		115.41 ± 7.70 ^a^	89.19 ± 1.89 ^d, e^	88.68 ± 2.38 ^d, e, f^	92.82 ± 2.59 ^c, d^	103.18 ± 1.06 ^b^	80.53 ± 5.79 ^f, g^	98.7 ± 1.31 ^b, c^	73.21 ± 2.79 ^g^	83.68 ± 5.43 ^e, f^
4	1-Hydroxy-2-propanone		131.49 ± 6.64 ^e^	154.86 ± 3.78 ^c^	144.97 ± 0.98 ^d^	167.15 ± 1.03 ^b^	193.62 ± 5.02 ^a^	150.96 ± 3.03 ^cd^	157.11 ± 6.42 ^c^	119.97 ± 6.75 ^f^	127.01 ± 2.74 ^e, f^
**Ketones**	**Subtotal**		1699.27	1085.24	968.49	1032.78	1351.48	1561.21	1715.09	1118.91	1030.06
	**Percentage**		2.57%	1.56%	1.38%	1.51%	1.91%	2.26%	2.42%	1.74%	1.54%
1	2-Methylpropanal	Pungent odor	30.59 ± 1.15 ^f^	60.48 ± 0.41 ^b^	52.04 ± 0.62 ^c^	40.2 ± 0.99 ^d^	64.01 ± 1.66 ^a^	32.19 ± 0.57 ^ef^	41.1 ± 3.31 ^d^	25.19 ± 0.56 ^g^	34.29 ± 1.49 ^e^
2	Propanal-M		119.46 ± 29.61 ^a^	78.61 ± 0.81 ^b^	77 ± 2.68 ^b^	70.94 ± 0.43 ^b^	123.98 ± 12.54 ^a^	73.21 ± 0.36 ^b^	113.58 ± 16.29 ^a^	85.86 ± 1.20 ^b^	76.04 ± 1.34 ^b^
3	Propanal-D		70.89 ± 28.49 ^a^	34.77 ± 0.72 ^d^	43.08 ± 2.16 ^c, d^	41.63 ± 1.56 ^c, d^	66.61 ± 10.15 ^a, b^	49.13 ± 4.08 ^a, b, c, d^	46.83 ± 12.25 ^b, c, d^	58.45 ± 2.62 ^a, b, c^	51.27 ± 0.52 ^a, b, c, d^
4	Acetaldehyde	Fruity, coffee aroma	1937.65 ± 475.32 ^a, b^	1968.56 ± 28.65 ^a, b^	1879.14 ± 22.19 ^a, b^	1957.42 ± 21.52 ^a, b^	2179.87 ± 170.56 ^a^	1941.76 ± 33.16 ^a, b^	2133.78 ± 205.86 ^a, b^	1751.3 ± 16.99 ^b^	1864.54 ± 33.56 ^a, b^
5	2-Methylbutanal	Stimulating, chocolatey	60.32 ± 1.68 ^a^	29.19 ± 2.38 ^d^	31.73 ± 0.22 ^d^	29.24 ± 0.64 ^d^	40.21 ± 2.35 ^b, c^	40.44 ± 2.02 ^bc^	44.02 ± 1.25 ^b^	39.95 ± 1.40 ^c^	39.75 ± 2.63 ^c^
6	3-Methylbutanal	Apple Scent	22.26 ± 1.65 ^g^	84.87 ± 1.16 ^a^	63.59 ± 0.74 ^c^	76.37 ± 0.87 ^b^	45.89 ± 0.89 ^d^	39.61 ± 1.97 ^e^	65.12 ± 0.44 ^c^	33.24 ± 1.10 ^f^	35.86 ± 2.20 ^f^
7	Pentanal	Pungent odor	126.14 ± 6.45 ^a^	61.69 ± 1.85 ^d^	57.87 ± 1.66 ^d^	45.71 ± 0.85 ^e^	85.07 ± 0.52 ^b^	71.37 ± 2.20 ^c^	75.55 ± 0.78 ^c^	70.98 ± 2.09 ^c^	49.46 ± 2.15 ^e^
8	Benzaldehyde	Bitter almond, cherry, nutty aroma	68.81 ± 4.51 ^a^	35.84 ± 0.52 ^d^	59.19 ± 1.93 ^b^	49.39 ± 4.36 ^c^	40.27 ± 6.08 ^c, d^	48.03 ± 4.93 ^c^	73.65 ± 5.17 ^a^	47.81 ± 6.73 ^c^	38.19 ± 3.10 ^d^
**Aldehydes**	**Subtotal**		2436.12	2354.01	2263.64	2310.9	2645.91	2295.74	2593.63	2112.78	2189.4
	**Percentage**		3.68%	3.37%	3.23%	3.37%	3.75%	3.32%	3.65%	3.29%	3.27%
1	Ethyl hexanoate-M	Currant, pineapple aroma	120.68 ± 7.68 ^e^	128.09 ± 0.46 ^c, d, e^	139.55 ± 3.89 ^b^	123.31 ± 2.49 ^d, e^	106.12 ± 0.41 ^f^	130.75 ± 6.86 ^b, c, d^	181.84 ± 5.76 ^a^	139.11 ± 3.07 ^b^	136.07 ± 0.90 ^b, c^
2	Isoamyl acetate-M		165.22 ± 12.28 ^d^	226.87 ± 7.65 ^a^	216.88 ± 17.25 ^a, b^	217.1 ± 2.93 ^a, b^	184.85 ± 18.82 ^c, d^	189.96 ± 10.19 ^c^	223.1 ± 3.45 ^a^	196.68 ± 2.20 ^b, c^	196.31 ± 2.81 ^b, c^
3	Ethyl 3-methylbutanoate-M	Fragrant and fruity with aromas of wine	129.41 ± 13.88 ^e^	193.65 ± 10.56 ^b, c^	150.77 ± 5.84 ^d^	214.36 ± 2.11 ^a^	176.46 ± 3.01 ^c^	181.36 ± 19.43 ^c^	203.01 ± 4.04 ^a, b^	86.33 ± 10.13 ^f^	86.35 ± 6.43 ^f^
4	Ethyl 3-methylbutanoate-D	Fragrant and fruity with aromas of wine	21.57 ± 2.67 ^e^	95 ± 5.08 ^b^	56.12 ± 4.63 ^d^	121.4 ± 2.46 ^a^	68.08 ± 0.80 ^c^	71.79 ± 2.46 ^c^	69.68 ± 1.28 ^c^	13.65 ± 0.72 ^f^	18.95 ± 0.82 ^e, f^
5	Isoamyl acetate-D	Banana odor	1890.69 ± 68.65 ^f^	4244.71 ± 131.95 ^d^	5204.44 ± 149.57 ^b^	4833.19 ± 23.81 ^c^	5168.42 ± 53.54 ^b^	4176.63 ± 98.52 ^d^	5264.53 ± 5.76 ^b^	3254.9 ± 85.74 ^e^	5813.75 ± 112.82 ^a^
6	Ethyl butanoate-D	Sour fruit, banana and strawberry flavors, floral and fruity aromas	631.01 ± 12.13 ^g^	785.32 ± 30.82 ^f^	955.85 ± 32.11 ^d^	845.43 ± 12.87 ^e^	1190.59 ± 11.08 ^b^	1075.47 ± 23.48 ^c^	1328.06 ± 9.43 ^a^	1201.34 ± 20.78 ^b^	1049.94 ± 26.47 ^c^
7	Isobutyl acetate-D	Ripe fruit aroma	540.83 ± 12.29 ^f^	589.71 ± 18.34 ^d^	658.25 ± 15.16 ^c^	330.02 ± 4.15 ^i^	1472.49 ± 5.54 ^a^	566.8 ± 9.54 ^e^	810.33 ± 1.95 ^b^	513.2 ± 11.46 ^g^	490.21 ± 9.16 ^h^
8	Propyl acetate	Fruity scent	141.04 ± 1.59 ^a^	79.25 ± 1.92 ^e, f^	75.6 ± 1.43 ^f^	55.65 ± 1.17 ^h^	119.18 ± 2.83 ^b^	82.71 ± 1.87 ^d, e^	84.52 ± 1.67 ^d^	99.86 ± 0.99 ^c^	63.08 ± 2.40 ^g^
9	Ethyl isobutyrate		325.2 ± 6.02 ^f^	1172.6 ± 16.99 ^a^	955.3 ± 3.57 ^c^	955.2 ± 7.34 ^c^	1079.03 ± 5.25 ^b^	726.55 ± 14.97 ^d^	513.88 ± 1.89 ^e^	126.34 ± 0.52 ^h^	225.32 ± 1.63 ^g^
10	Ethyl propanoate	Pineapple scent	372.18 ± 11.97 ^e^	358.38 ± 4.07 ^e, f^	382.63 ± 14.51 ^e^	410.04 ± 4.15 ^c^	444.94 ± 6.58 ^b^	368.5 ± 25.00 ^d, e, f^	537.94 ± 6.57 ^a^	343.44 ± 11.57 ^f^	400.61 ± 12.72 ^c, d^
11	Ethyl Acetate	Sweet and fruity	10,418.17 ± 58.61 ^a^	8558.08 ± 29.86 ^g^	8580.84 ± 32.12 ^g^	8666.27 ± 26.45 ^f^	9466.89 ± 24.73 ^b^	8695.02 ± 1.68 ^f^	8820.4 ± 25.45 ^e^	8984.57 ± 44.27 ^d^	9151.87 ± 29.06 ^c^
12	Methyl acetate		1770.32 ± 18.12 ^c^	1876.34 ± 13.17 ^b^	1709.77 ± 2.66 ^e^	1634.92 ± 3.14 ^f^	1424.28 ± 5.12 ^g^	1747.63 ± 2.77 ^d^	1952.51 ± 1.96 ^a^	806.49 ± 9.20 ^i^	871.49 ± 2.42 ^h^
13	Ethyl hexanoate-D	Currant and pineapple aroma	175.03 ± 8.00 ^h^	484.28 ± 33.10 ^c, d^	531.94 ± 41.84 ^c^	401.35 ± 9.05 ^e, f^	432.93 ± 14.47 ^d, e^	678.16 ± 19.57 ^b^	1033.31 ± 28.26 ^a^	302.95 ± 19.12 ^g^	354.26 ± 34.39 ^f, g^
14	Methyl 3-methylbutanoate		435.08 ± 3.61 ^a^	272.37 ± 3.18 ^f^	306.73 ± 6.35 ^e^	276.45 ± 3.92 ^f^	349.16 ± 4.65 ^c^	314.83 ± 11.15 ^d, e^	371.53 ± 18.05 ^b^	374.51 ± 7.22 ^b^	331.76 ± 6.06 ^d^
15	Ethyl octanoate	Fruity aromas with pineapple, apple-like notes and brandy wine notes	156.27 ± 2.14 ^i^	246.98 ± 6.90 ^f^	272.98 ± 14.03 ^d^	236.65 ± 6.81 ^g^	275.49 ± 8.84 ^c^	234.34 ± 26.46 ^h^	359 ± 8.31 ^a^	268.22 ± 4.52 ^e^	289.98 ± 20.82 ^b^
16	Isobutyl propionate	Rum and pineapple aromas	655.31 ± 8.26 ^a^	242.04 ± 5.79 ^f^	233.95 ± 1.55 ^f^	251.44 ± 3.93 ^f^	103.33 ± 3.32 ^g^	421.49 ± 17.66 ^d^	288.05 ± 3.79 ^e^	629.72 ± 12.81 ^b^	470.76 ± 9.78 ^c^
17	Hexyl acetate	Fruity scent	22.95 ± 2.40 ^e^	41.3 ± 2.55 ^d^	62.63 ± 4.27 ^b^	39.07 ± 1.84 ^d^	34.5 ± 0.57 ^d^	59.1 ± 5.90 ^b, c^	96.94 ± 1.31 ^a^	34.09 ± 4.90 ^d^	53.86 ± 5.31 ^c^
18	Ethyl lactate	Pungent odor	491.84 ± 47.44 ^b, c^	422.79 ± 6.86 ^d^	452.63 ± 28.74 ^c, d^	486.03 ± 5.84 ^b, c^	433.68 ± 22.02 ^d^	486.22 ± 21.90 ^b, c^	268.27 ± 29.62 ^e^	533.47 ± 4.14 ^a, b^	561.66 ± 7.30 ^a^
19	Isobutyl butyrate	Pineapple, grape skin aromas, and etheric notes	117.43 ± 5.33 ^e^	196.12 ± 9.54 ^c^	196.16 ± 9.10 ^c^	232.18 ± 19.26 ^b^	145.69 ± 3.73 ^d^	187 ± 8.19 ^c^	235.69 ± 5.18 ^b^	259.67 ± 12.07 ^a^	247.36 ± 15.40 ^a, b^
20	Ethyl 2-methylbutanoate	Aromas of apple skin, pineapple skin, and unripe plum skin	27.55 ± 1.55 ^f, g^	113.03 ± 6.01 ^b^	83.84 ± 3.15 ^d^	168.06 ± 1.68 ^a^	55.25 ± 1.71 ^e^	85.27 ± 1.63 ^c, d^	89.62 ± 0.36 ^c^	23.32 ± 0.84 ^g^	29.4 ± 1.65 ^f^
**Esters**	**Subtotal**		18,607.78	20,326.91	21,226.86	20,498.12	22,731.36	20,479.58	22,732.21	18,191.86	20,842.99
	**Percentage**		28.14%	29.13%	30.29%	29.88%	32.21%	29.65%	32.03%	28.29%	31.16%
1	Acetic acid-M	Sour and pungent odor in vinegar	2377.63 ± 155.71 ^b^	2390.38 ± 32.60 ^b^	2444.41 ± 37.53 ^a, b^	2540.6 ± 14.61 ^a, b^	2162.19 ± 101.49 ^c^	2409.18 ± 48.41 ^b^	2056.72 ± 215.15 ^c^	2510.27 ± 20.88 ^a, b^	2612.47 ± 5.78 ^a^
2	Acetic acid-D	Sour and pungent odor in vinegar	166.65 ± 33.73 ^c^	169.32 ± 4.25 ^c^	176.75 ± 6.41 ^c^	209.31 ± 8.56 ^a, b^	79.75 ± 13.71 ^e^	120.48 ± 8.51 ^d^	100.68 ± 23.02 ^d, e^	179.06 ± 8.49 ^b, c^	222.12 ± 2.00 ^a^
**Acids**	**Subtotal**		2544.28	2559.7	2621.16	2749.91	2241.94	2529.66	2157.4	2689.33	2834.59
	**Percentage**		3.85%	3.67%	3.74%	4.01%	3.18%	3.66%	3.04%	4.18%	4.24%
1	Terpinolene	Resinous pine wood scent	48.02 ± 6.56 ^a^	27.67 ± 3.35 ^b, c^	19.57 ± 1.41 ^d^	23.58 ± 0.76 ^b, c, d^	30.03 ± 2.24 ^b^	22.26 ± 3.45 ^c, d^	29.74 ± 5.98 ^b^	24.99 ± 1.99 ^b, c, d^	25.41 ± 0.87 ^b, c, d^
**Terpenoids**	**Subtotal**		48.02	27.67	19.57	23.58	30.03	22.26	29.74	24.99	25.41
	**Percentage**		0.07%	0.04%	0.03%	0.03%	0.04%	0.03%	0.04%	0.04%	0.04%
1	2,5-Dimethylfuran	Pungent aroma of fried flowers and the smell of chocolate and cream	199.03 ± 4.17 ^f^	662.49 ± 11.18 ^a^	289.7 ± 6.39 ^d^	235.49 ± 8.54 ^e^	194.62 ± 3.32 ^f^	551.15 ± 10.22 ^b^	341.04 ± 6.08 ^c^	159.62 ± 2.09 ^g^	117.35 ± 5.38 ^h^
**Furans**	**Subtotal**		199.03	662.49	289.7	235.49	194.62	551.15	341.04	159.62	117.35
	**Percentage**		0	0.01	0	0	0	0.01	0	0	0
	**Total**		58,160.24	42,201.41	34,187.72	37,576.72	53,287.78	39,846.59	43,863.56	43,678.17	52,144.75

Means with different letters in the same column express significant differences (Duncan’s test *p* < 0.05). _M and _D, which are the Monomer and Dimer of the same substance.

**Table 7 foods-11-02767-t007:** OAV Analysis of Main Aroma Compounds in Different Variety Wine Samples.

No.	Substance Name	‘Hassan’	‘Zuoshaner’	‘Beibinghong’	‘Zuoyouhong’	‘Beta’	‘Shuanghong’	‘Zijingganlu’	‘Cabernet Sauvignon’	‘Syrah’
(Aldehydes)1	2-Methylpropanal	33.99 ± 1.28	67.2 ± 0.46	57.82 ± 0.69	44.67 ± 1.1	71.12 ± 1.84	35.77 ± 0.63	45.66 ± 3.68	27.99 ± 0.62	38.1 ± 1.65
2	Acetaldehyde	3.88 ± 0.95	3.94 ± 0.06	3.76 ± 0.04	3.91 ± 0.04	4.36 ± 0.34	3.88 ± 0.07	4.27 ± 0.41	3.5 ± 0.03	3.73 ± 0.07
3	2-Methylbutanal	46.4 ± 1.29	22.46 ± 1.83	24.41 ± 0.17	22.49 ± 0.49	30.93 ± 1.8	31.11 ± 1.55	33.86 ± 0.96	30.73 ± 1.07	30.57 ± 2.02
(Esters)1	Ethyl hexanoate-M	8.62 ± 0.55	9.15 ± 0.03	9.97 ± 0.28	8.81 ± 0.18	7.58 ± 0.03	9.34 ± 0.49	12.99 ± 0.41	9.94 ± 0.22	9.72 ± 0.06
2	Isoamyl acetate-M	55.07 ± 4.09	75.62 ± 2.55	72.29 ± 5.75	72.37 ± 0.97	61.62 ± 6.27	63.32 ± 3.4	74.37 ± 1.15	65.56 ± 0.74	65.44 ± 0.94
3	Ethyl 3-methylbutanoate-M	12.94 ± 1.39	19.36 ± 1.06	15.08 ± 0.58	21.44 ± 0.21	17.65 ± 0.3	18.14 ± 1.94	20.3 ± 0.4	8.63 ± 1.01	8.63 ± 0.64
4	Ethyl 3-methylbutanoate-D	2.16 ± 0.27	9.5 ± 0.51	5.61 ± 0.46	12.14 ± 0.25	6.81 ± 0.08	7.18 ± 0.25	6.97 ± 0.13	1.37 ± 0.07	1.89 ± 0.08
5	Isoamyl acetate-D	20.11 ± 0.73	45.16 ± 1.40	55.37 ± 1.59	51.42 ± 0.25	54.98 ± 0.57	44.43 ± 1.05	56.01 ± 0.06	34.63 ± 0.91	61.85 ± 1.2
6	Ethyl butanoate-D	31.55 ± 0.61	39.27 ± 1.54	47.79 ± 1.61	42.27 ± 0.64	59.53 ± 0.55	53.77 ± 1.17	66.4 ± 0.47	60.07 ± 1.04	52.5 ± 1.32
7	Ethyl isobutyrate	16.26 ± 0.3	58.63 ± 0.85	47.76 ± 0.18	47.76 ± 0.37	53.95 ± 0.26	36.33 ± 0.75	25.69 ± 0.09	6.32 ± 0.03	11.27 ± 0.08
8	Ethyl Acetate	1.39 ± 0.01	1.14 ± 0.00	1.14 ± 0.00	1.16 ± 0	1.26 ± 0	1.16 ± 0	1.18 ± 0	1.2 ± 0.01	1.22 ± 0
9	Ethyl hexanoate-D	12.5 ± 0.57	34.59 ± 2.36	38 ± 2.99	28.67 ± 0.65	30.92 ± 1.03	48.44 ± 1.4	73.81 ± 2.02	21.64 ± 1.37	25.3 ± 2.46
10	Ethyl octanoate	31.25 ± 0.43	49.4 ± 1.38	54.6 ± 2.81	47.33 ± 1.36	55.1 ± 1.77	46.87 ± 5.29	71.8 ± 1.66	53.64 ± 0.9	58 ± 4.16
(Furans) 1	2,5-Dimethylfuran	2.49 ± 0.05	8.28 ± 0.14	3.62 ± 0.08	2.94 ± 0.11	2.43 ± 0.04	6.89 ± 0.13	4.26 ± 0.08	2 ± 0.03	1.47 ± 0.07

**Table 8 foods-11-02767-t008:** Analysis of VIP Values of Aroma Compounds in Different Varieties of Wine.

Compound Name	‘Hassan’ vs. ‘Zuoshaner’ vs. ‘Beibinghong’ vs. ‘Zuoyouhong’ vs. ‘Beta’ vs. ‘Shuanghong’ vs. ‘Zijingganlu’ vs. ‘Cabernet Sauvignon’ vs. ‘Syrah’
Ethyl isobutyrate	1.76542
Ethyl hexanoate-D	1.46159
2-Methylpropanal	1.37144
Ethyl octanoate	1.29694
Ethyl butanoate-D	1.24456
Isoamyl acetate-D	1.15269

## Data Availability

All related data and methods are presented in this paper. Additional inquiries should be addressed to the corresponding author.

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
