# Peer review of "Characterization of the Key Aroma Volatile Compounds in Nine Different Grape Varieties Wine by Headspace Gas Chromatography–Ion Mobility Spectrometry (HS-GC-IMS), Odor Activity Values (OAV) and Sensory Analysis"

_foods, 2022, doi:10.3390/foods11182767_

Round 1
Reviewer 1 Report
I suggest the publication of the work after major revision. In my opinion, more work is needed in the identification of the compounds. In the section 3.5 authors described the identification of volatiles based on the comparison with NIST and IMS database. I suggest calculating retention indices by using N-ketones, and comparing them with the drift time of the standard in the GC–IMS library.
I suggest to authors to add more detail about data analysis and model building in the materials and methods section. How were the data normalized for baseline correction? In my opinion, it is not enough to mention this only under the figure. Have the authors used some scaling methods for the data normalization? Etc.
Other minor observations:
Introduction
Line 80: add citations about previous papers about wine
Lines 82, 83: add citations (Firstly, previous studies have only analyzed the aroma components in wine, failing to identify the key aroma components)
Lines 84, 85: add citations (Secondly, many previous studies have reported the application of HS-GC-IMS for detecting volatiles in food and agricultural products)
Reviewer 2 Report
In the present manuscript, the physicochemical characteristics, color, volatile compounds, and sensory features of the original wine made from nine varieties of grapes in China were studied. The article is well designed and has enough experiments. The results are well presented and discussed. In my opinion, the present manuscript can be published in Foods after minor revisions as follows:
L13-14: Delete "and sensory evaluation".
L22: Change "beta" to "Beta".
L23: Write "odor activity value" before OAV.
L25: OVA is not correct.
L25: What is VIP?
L31: Do not use abbreviations in the keywords.
L45-51: This section needs Reference/s.
L75-79: This section needs Reference/s.
L80-89: This section needs Reference/s.
L145: How were the grapes crushed? By hand or a special tool?
L145-149: Explain more about fermentation conditions. Type and size of fermentation containers? What volume of containers was filled with crushed grapes? Was the lid of the container tightly closed?
-How was the SO2 added to the samples?
-How was the completion time of first and secondary fermentation determined?
Section 2.3.2. needs a reference for analysis methods.
L190: Is 4-methyl-2-pentanol internal standard?
L208: How was the odor threshold measured? Explain its measurement method.
L303: change "the" to "The".
Table 6. Write the data in a smaller size. Also, check the categories' compounds again.
-Was there a significant relationship between the sensory evaluation results and the main aroma compounds of the samples?
Reviewer 3 Report
The paper is interesting but some improvements should be included. Discussion could be improved by creating the connections within different groups of the results and explanations from the literature. Beside that, minor changes are following:
Line 146 please indicate the yeast name. Winemaking process should be described in more details. How long was the first fermentation, what was the temperature, etc.
Table 3 reduce the letter size in order to arrange the look of the table.
Lines 246-248 How can you explain the extreme differences in anthocyanins content for example in ‘Zuoshaner’ and Beta grape varieties.
Correct P value as small letter and italic p
Line 263 Why is chromaticity important i.e. what was the reason for determination of absorbance and how the differences affect wine characteristics?
Line 278 the colour should be discussed from point of anthocyanins content
Lines 326-332 How can you explain such a significant difference of Beta grape variety from the other ones?
Table 6 Please indicate the references for the aroma descriptors and the concentration unit in the table
Table 6 is very uncomprehensive and hard to read so it must be rearranged
Line 408 Lipids can not be the aroma component
Table 7 Indicate the OT values with appropriate references
Lines 578-583 Sensory attributes should be explained in more detail regarding the GC-MS results of wine samples.
The discussion part was involved within the results, while Conclusion and discussion contains only conclusions.
Round 2
Reviewer 1 Report
I accept the revised manusript, in my opinion, it can be published in this form